# Adaptation of the periplasm to maintain spatial constraints essential for cell envelope processes and cell viability

Eric Mandela[1†], Christopher J Stubenrauch[1†], David Ryoo[2], Hyea Hwang[3‡], Eli J Cohen[4], Von L Torres[1], Pankaj Deo[1], Chaille T Webb[1], Cheng Huang[5], Ralf B Schittenhelm[5], Morgan Beeby[4], JC Gumbart[6*], Trevor Lithgow[1*], Iain D Hay[7*]

[1]Infection & Immunity Program, Biomedicine Discovery Institute and Department of Microbiology, Monash University, Clayton, Australia; [2]Interdisciplinary Bioengineering Graduate Program, Georgia Institute of Technology, Atlanta, United States; [3]School of Materials Science and Engineering, Georgia Institute of Technology, Atlanta, United States; [4]Department of Life Sciences, Imperial College London, London, United Kingdom; [5]Monash Proteomics & Metabolomics Facility, Department of Biochemistry and Molecular Biology, Biomedicine Discovery Institute, Monash University, Clayton, Australia; [6]School of Physics, Georgia Institute of Technology, Atlanta, United States; [7]School of Biological Sciences, The University of Auckland, Auckland, New Zealand

**\*For correspondence:**
gumbart@physics.gatech.edu (JCG);
trevor.lithgow@monash.edu (TL);
iain.hay@auckland.ac.nz (IDH)

[†]These authors contributed equally to this work

**Present address:** [‡]NIH Center for Macromolecular Modeling and Bioinformatics, Beckman Institute for Advanced Science and Technology, Department of Biochemistry, Center for Biophysics and Quantitative Biology, University of Illinois at Urbana-Champaign, Urbana, United States

**Competing interest:** The authors declare that no competing interests exist.

**Abstract** The cell envelope of Gram-negative bacteria consists of two membranes surrounding a periplasm and peptidoglycan layer. Molecular machines spanning the cell envelope depend on spatial constraints and load-bearing forces across the cell envelope and surface. The mechanisms dictating spatial constraints across the cell envelope remain incompletely defined. In *Escherichia coli*, the coiled-coil lipoprotein Lpp contributes the only covalent linkage between the outer membrane and the underlying peptidoglycan layer. Using proteomics, molecular dynamics, and a synthetic lethal screen, we show that lengthening Lpp to the upper limit does not change the spatial constraint but is accommodated by other factors which thereby become essential for viability. Our findings demonstrate *E. coli* expressing elongated Lpp does not simply enlarge the periplasm in response, but the bacteria accommodate by a combination of tilting Lpp and reducing the amount of the covalent bridge. By genetic screening, we identified all of the genes in *E. coli* that become essential in order to enact this adaptation, and by quantitative proteomics discovered that very few proteins need to be up- or down-regulated in steady-state levels in order to accommodate the longer Lpp. We observed increased levels of factors determining cell stiffness, a decrease in membrane integrity, an increased membrane vesiculation and a dependance on otherwise non-essential tethers to maintain lipid transport and peptidoglycan biosynthesis. Further this has implications for understanding how spatial constraint across the envelope controls processes such as flagellum-driven motility, cellular signaling, and protein translocation

## Editor's evaluation

In this study, Mandela et al. investigate the response of cells to a lengthened version of the periplasmic protein Lpp (Lpp[+21]). An abundant protein present in high abundance in cells, Lpp tethers the outer membrane to the peptidoglycan layer and is implicated in maintaining the distance between the two. Combining genetics, proteomics and simulations, the authors determine that

lengthening Lpp does not change the spatial organization of the periplasm due to apparently compensatory effects. Together, these data highlight the importance of periplasmic organization to cell viability and the resilience of the systems that maintain it.

## Introduction

Gram-negative bacteria have a cell envelope composed of two membranes sandwiching between them an aqueous space called the periplasm, in which an essential structural layer of peptidoglycan (PG) resides. The outer membrane is critical to cell growth and these bacteria face challenges to their cell biology in terms of membrane protein assembly and lipid-transport pathways that must traverse the distance from the inner membrane (IM) to the outer membrane (OM) (*Silhavy et al., 2010*). Recent work investigating the spatial demands for assembly of proteins into the outer membrane has shown precincts of active protein integration into the membrane can deliver new material to the growing outer membrane (*Gunasinghe et al., 2018*) and that random planar movement from these precincts drives the observed non-uniform distributions of the major proteins of the outer membrane (*Rassam et al., 2015*; *Ursell et al., 2012*). By contrast to these protein components that diffuse to the outer membrane, elements of the LPS-transport machinery (e.g. the Lpt complex) (*Freinkman et al., 2012*; *Sperandeo et al., 2017*; *Sperandeo et al., 2011*; *Ekiert et al., 2017*; *Shrivastava and Chng, 2019*) span the OM and IM in order to fulfil their function in delivering lipid components to the outer membrane.

The PG layer is a fundamental aspect of the cell envelope, and it must be dynamically remodeled to allow growth as well as the assembly and transit of trans-envelope structures. PG synthesis and remodeling is a complex process with high levels of redundancy at various steps, involving at least 50 enzymes in *E. coli* (*Pazos et al., 2017*). The Penicillin-binding proteins (PBPs) are the core components responsible for the periplasmic biosynthesis of peptidoglycan. There are multiple PBP complexes including the two core, semi-redundant PBP complexes PBP1a and PBP1b embedded in the IM which are activated by interactions with lipoproteins LpoA and LpoB embedded in the OM. Thus, the activation of PG synthesis by these enzymes is spatially regulated, serving as a self-limiting molecular ruler to modulate PG thickness (*Typas et al., 2011*). Cells must possess either a functional PBP1a or PBP1b system for growth (*Paradis-Bleau et al., 2010*; *Typas et al., 2010*).

Trans-envelope complexes such as lipopolysaccharide (LPS) transit pathways (*Freinkman et al., 2012*; *Sperandeo et al., 2017*; *Sperandeo et al., 2011*; *Ekiert et al., 2017*; *Shrivastava and Chng, 2019*) and the protein translocation and assembly module (the TAM) (*Selkrig et al., 2015*; *Selkrig et al., 2012*; *Shen et al., 2014*) are also spatially constrained by the need to reach across from the IM to OM in order to function. The PG layer is covalently attached to the outer membrane by Braun's lipoprotein (Lpp), with two recent papers addressing whether extending the length of Lpp would impact on (i) flagellar function given that the flagellum spans both membranes (*Cohen et al., 2017*) and (ii) signal-transduction systems that span the OM to IM (*Asmar et al., 2017*). In both cases, extending the coiled-coil structure of Lpp by 21 residues (Lpp[+21]) was found to be the longest form that supported close to normal growth (*Cohen et al., 2017*; *Asmar et al., 2017*). Imaging of these Lpp[+21] strains in both studies showed that the total periplasmic width had been stretched ~3–4 nm (*Cohen et al., 2017*; *Asmar et al., 2017*). This being the case, the Lpp[+21] model would provide a powerful experimental system to study how processes like OM biogenesis and PG biosynthesis can be maintained under a spatial stress on the cell wall.

To understand how trans-envelope processes in *E. coli* adapt to the presence of an enlarged periplasm, a combination of phenotypic analysis, proteomics, molecular dynamics and a synthetic lethal screen was employed to identify and characterize factors needed to maintain viability in the Lpp[+21] strain of *E. coli*. The genetic screen demanded synthetic growth phenotypes from an array of mutants each lacking a gene that, while non-essential in wild-type *E. coli* (*Baba et al., 2006*), is essential in the Lpp[+21] strain. These genes fall into three functional categories: PG biosynthesis and remodeling, LPS biosynthesis and PG-outer membrane linkage. We show that previously non-essential proteins involved in bridging the gap between the OM and PG become essential in the context of the Lpp[+21] strain background. These include previously known PG binding OM proteins (OmpA and Pal) as well as proteins previously not known to play an active role in linking the OM and PG (TolC and YiaD). We observed a thicker more diffuse or heterogeneous PG layer in the Lpp[+21] strain and whole cell

proteomics revealed that in response to an increased length of Lpp, *E. coli* increases the levels of a range of cell envelope proteins involved in PG turnover. We discuss the outcomes in terms of how the PG-outer membrane linkage functionalizes the periplasm, the evolutionary constrains in place to maintain this functionality, and the specific activity of Lpp in contributing to the load-bearing function of the OM.

## Results

### Resilience and growth of the Lpp$^{+21}$ strain

A phylogenetic assessment of Lpp lengths across diverse bacterial lineages showed a very narrow window of protein size (*Figure 1A*), with Lpp being 78 residues in most species of bacteria including *E. coli*. Lpp lengths of 99 residues or more are at the upper end of the natural range for this protein and, in nature, these longer Lpp proteins are found in the genus *Geobacter*. The previously described Lpp$^{+21}$ isoform expressed in *E. coli* therefore sits near the upper physiological range observed among bacteria. We introduced a gene encoding the Lpp$^{+21}$ isoform into an *E. coli* background suitable for genetic screens (*Figure 1—figure supplement 1*) and confirmed the size of the protein by SDS-PAGE of bacterial cell extracts (*Figure 1B*). Enteric bacteria like *E. coli* can thrive in hyperosmotic environments compared to most laboratory growth media, such as the human gut by maintaining the periplasm and the cytoplasm in an iso-osmotic state (*Ingraham and Marr, 1996*; *Stock et al., 1977*). This is achieved by adjusting the solute concentration in the cell compartments by influx or efflux of water. By doing so, osmolality contributes immensely to the architectural aspects of cellular compartments (*Stock et al., 1977*; *Cayley et al., 2000*). In this study sorbitol was used to mimic these physiological osmotic conditions. Previous studies have demonstrated that the periplasmic volume increased rapidly in response to increased osmolyte levels in the external medium. This phenomenon can reduce the cytoplasmic volume by about 30%, thereby constricting the IM inwards, and substantially increasing the periplasmic volume by around 300% (*Stock et al., 1977*; *Cayley et al., 2000*; *Pilizota and Shaevitz, 2013*). The Lpp$^{+21}$ isoform had little impact on growth of *E. coli*. On minimal growth medium, growth rates of the Lpp$^{+21}$ strain of *E. coli* were equivalent to the isogenic wild-type *E. coli* (*Figure 1C*). This was likewise true on growth media osmotically balanced with concentrations of sorbitol up to 1.0 M (*Figure 1D*), and on rich (LB) medium with or without sorbitol (*Figure 1D*).

To establish the extent to which the periplasm had been remodeled in the Lpp$^{+21}$ strain of *E. coli*, we compared the periplasms of WT and Lpp$^{+21}$ strains using electron cryotomography in which cells are preserved in a frozen-hydrated, near-native state. To discern peptidoglycan, which was indistinct in previous studies, we increased the signal to noise ratio in images by calculating subtomogram averages instead of inspecting individual tomograms. As expected, the average distance from the middle of the IM density to the middle of the OM density was increased in both strains under hyperosmotic conditions when compared to previously reported data where cells were grown in standard laboratory media (*Asmar et al., 2017*). As shown in *Figure 1E*, the averaged distance from the IM to the OM were somewhat greater in the Lpp$^{+21}$ strain (32–36 nm) compared to the isogenic wild-type strain (30–32 nm) in line with previous Lpp$^{+21}$ periplasmic width measurements. Previous studies could not discern peptidoglycan; whereas here, we were able discern peptidoglycan in subtomogram averages. The distance to the center of the PG density from the OM was slightly increased in the Lpp$^{+21}$ strain (*Figure 1F*), although by less than the anticipated ~3 nm. The PG morphology was also qualitatively different: in the wild-type strain a uniform dark PG layer could be observed in the images, whereas in the Lpp$^{+21}$ strain a broader and more diffuse PG layer was evident, suggesting heterogeneity in both the density and thickness of the PG.

### Cell envelope response to elongated Lpp

To determine the adaptive response to changing the distance constraint between OM and PG, quantitative whole cell proteomics was applied to evaluate the Lpp$^{+21}$ strain. Triplicate samples of the wild-type and Lpp$^{+21}$ strains were processed for analysis by mass spectrometry and we sought to identify those proteins where the steady-state level increases or decreases three-fold or more (Log$_2$ fold change of ±1.6) in the Lpp$^{+21}$ strain (*Figure 2A*; *Supplementary file 1*). The level of the oligopeptide transporter subunits (OppB, OppC, OppD and OppF) are substantially increased in the Lpp$^{+21}$ strain compared to the wild-type (*Supplementary file 2*). This suggests an increase in PG turnover and an

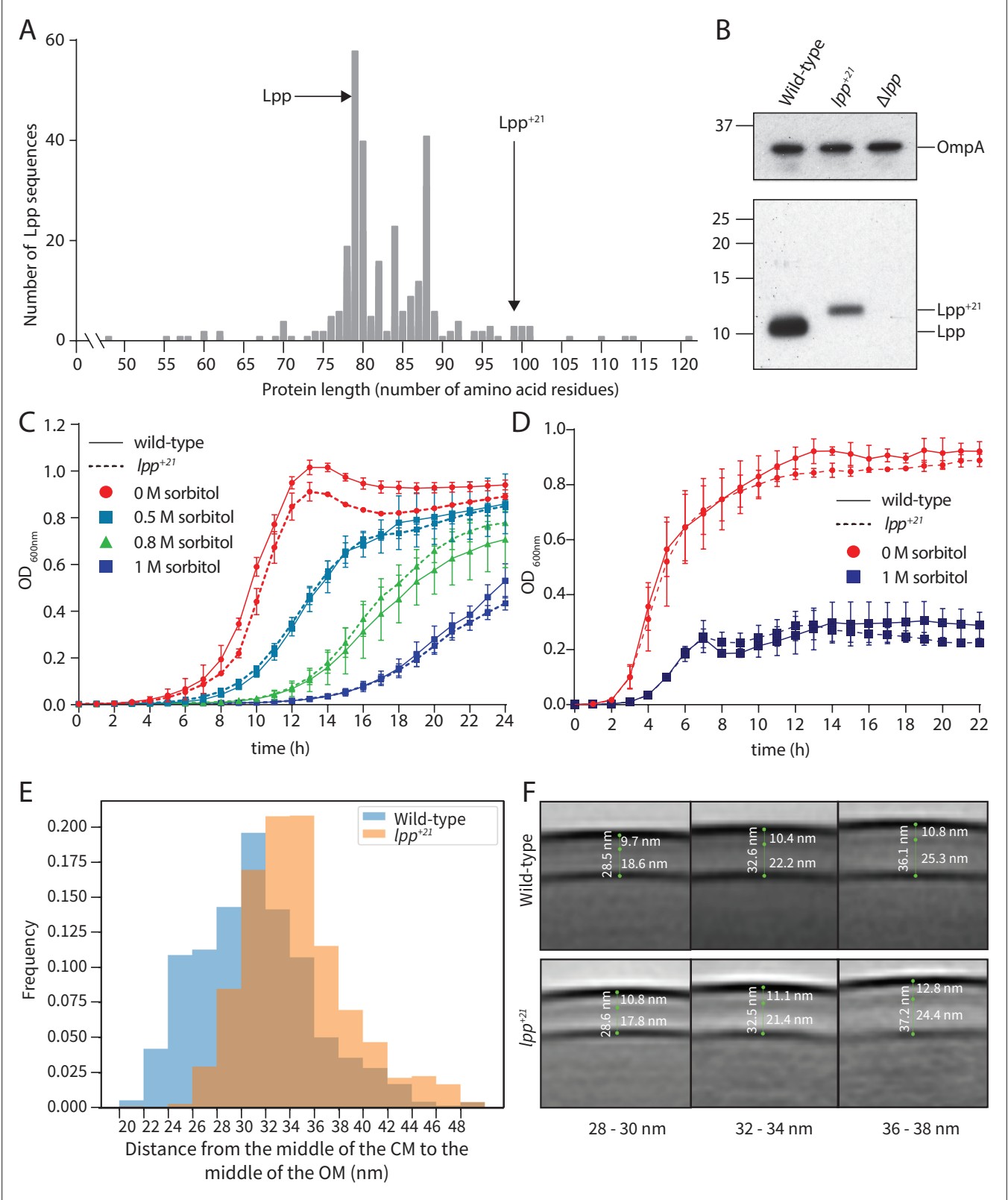

**Figure 1.** Phenotypes of *E. coli* cells encoding the Lpp+21 isoform. (**A**) Non-redundant Lpp sequences were identified (Materials and methods) and the protein length charted on the x-axis. The number of non-redundant sequences showing that length is shown on the y-axis. The location of Lpp and the lengthened Lpp+21 are indicated. (**B**) Whole cell lysates were prepared from the indicated strains and subject to SDS-PAGE and immunoblot analysis with anti-Lpp antibodies and anti-OmpA antibodies. OmpA serves as a loading control. (**C**) The JW5028 – Keio BW25113 strain with *kan* gene replacing

*Figure 1 continued on next page*

*Figure 1 continued*

a pseudogene background and isogenic Lpp$^{+21}$ strain (*Figure 1—figure supplement 1*) were grown over 24 hr. The growth medium is M9, containing the indicated concentration of sorbitol as an osmolyte. (**D**) Growth rates for the same strains were measured in rich (LB) growth media with and without sorbitol over 20 hr. (**E**) The periplasmic width distribution of the indicated strains in hyperosmotic conditions. The histogram depicts the frequency with which a given distance is observed between the OM and PG. (**F**) Subtomogram averages of cell envelopes in hyperosmotic conditions. Measurements from EM views evaluate the distance between OM and PG in the Lpp$^{+21}$ strain. While the PG in the wild-type strain is a uniform thin electron dense layer, the PG layer in the Lpp$^{+21}$ strain is more diffuse and thicker. Each panel represents averages of the subtomogram cell envelope section binned into the sizes shown.

The online version of this article includes the following figure supplement(s) for figure 1:

**Figure supplement 1.** Construction and assessment of Lpp$^{+21}$ *E. coli*.

overall increased capacity to recycle PG components and is consistent with the concomitant increase in AmiC, one of the two major amidases involved in PG remodeling. In addition, proteins implicated in diverse stress-responses (cold shock proteins CspG, CspA, CspI, and YdfK, as well as envelope stress protein ZraP and redox stress protein YfcG) were observed at increased steady-state levels in Lpp$^{+21}$ strain (*Figure 2A*, *Supplementary file 2*). The greatest decreases were seen in the steady state levels of the GatZABCD proteins involved in galactitol phosphotransferase system and DHAP synthesis (*Figure 2A*). The *gatABCD* genes have been shown to be responsive to factors that change *E. coli* cell surface tension (*Domka et al., 2007*) and Lpp$^{+21}$ has been reported to significantly decrease cell stiffness (*Mathelié-Guinlet et al., 2020*).

A decrease was seen in the steady-state level of the Lpp$^{+21}$ isoform in the mutant, to approximately one-eighth the level of Lpp in the wild-type strain (*Figure 2—figure supplement 2*). This is consistent with the relative abundance of Lpp and Lpp$^{+21}$ observed in SDS-PAGE analysis of cell extracts from the two strains (*Figure 1F*). However, despite the relative decrease, Lpp$^{+21}$ remains as a highly abundant component of the OM-PG linkage factors given that Lpp is present at up to $10^6$ protein molecules per wild-type cell (*Silhavy et al., 2010*; *Li et al., 2014*).

The mass spectrometry data was processed to allow for an analysis of sub-cellular proteomes (*Loos et al., 2019*; *Figure 2—figure supplement 2*). An initially puzzling observation was that the Lpp$^{+21}$ strain has a 12% overall reduction of total periplasmic protein compared to wild type (*Figure 2B*). This was calculated as the proportion of the summed intensity from identified proteins predicted to reside in the periplasm in the STEPdb: G, E, F2, F3, I annotations (*Loos et al., 2019*; *Figure 2—figure supplement 2*). Both Lpp$^{+21}$ strains and null Lpp strains of *E. coli* are softer as previously adjudged by atomic force microscopy (*Mathelié-Guinlet et al., 2020*), and several factors that increase the softness of *E. coli* are also implicated in an increased outer membrane vesicle (OMV) production (*Cohen et al., 2017*; *Sonntag et al., 1978*; *Schwechheimer et al., 2014*; *Rojas et al., 2017*). To address whether the measured depletion of periplasmic content reflects an increased production of OMVs, extracts measuring the amount of total protein in the OMV fraction were normalized to OD $_{600}$ (*Figure 2C*, *Figure 2—figure supplement 2*). This confirmed that the presence of Lpp$^{+21}$ promotes approximately 10-fold more total protein associated with the OMV fraction, reflecting increased OMV production. The overall level of OM proteins associated with the cells was maintained constant (Figure S3) but a small OM integrity defect was evident from an increased sensitivity to SDS (*Figure 2D*).

## Lpp$^{+21}$ can be accommodated in the periplasm, but other factors become essential

To directly address the altered phenotype induced by Lpp$^{+21}$, we established a synthetic genetic array for factors in *E. coli* that become essential in order to maintain viability of the Lpp$^{+21}$ strain (*Figure 3—figure supplement 1*). The screen demands synthetic growth phenotypes from an array of mutants each lacking a single gene that, while non-essential in wild-type *E. coli*, become essential in the Lpp$^{+21}$ strain of *E. coli*. The endogenous *lpp* gene was replaced by a gene encoding Lpp$^{+21}$ in a isogenic library of 3818 *E. coli* mutants, each of which lacks a non-essential gene. Growth on rich medium allowed the rescue of the new library for array into a format suitable for high-throughput screening with a Singer RoToR robotics platform (*Figure 3—figure supplement 1*). Phenotypic analysis was thereafter scored for growth by comparing the growth of the isogenic mutants in the Lpp background (*Figure 3A*) with the equivalent mutants in the Lpp$^{+21}$ background (*Figure 3B*). Each of the genes that displayed a noticeable phenotype in these analyses are presented in *Table 1*.

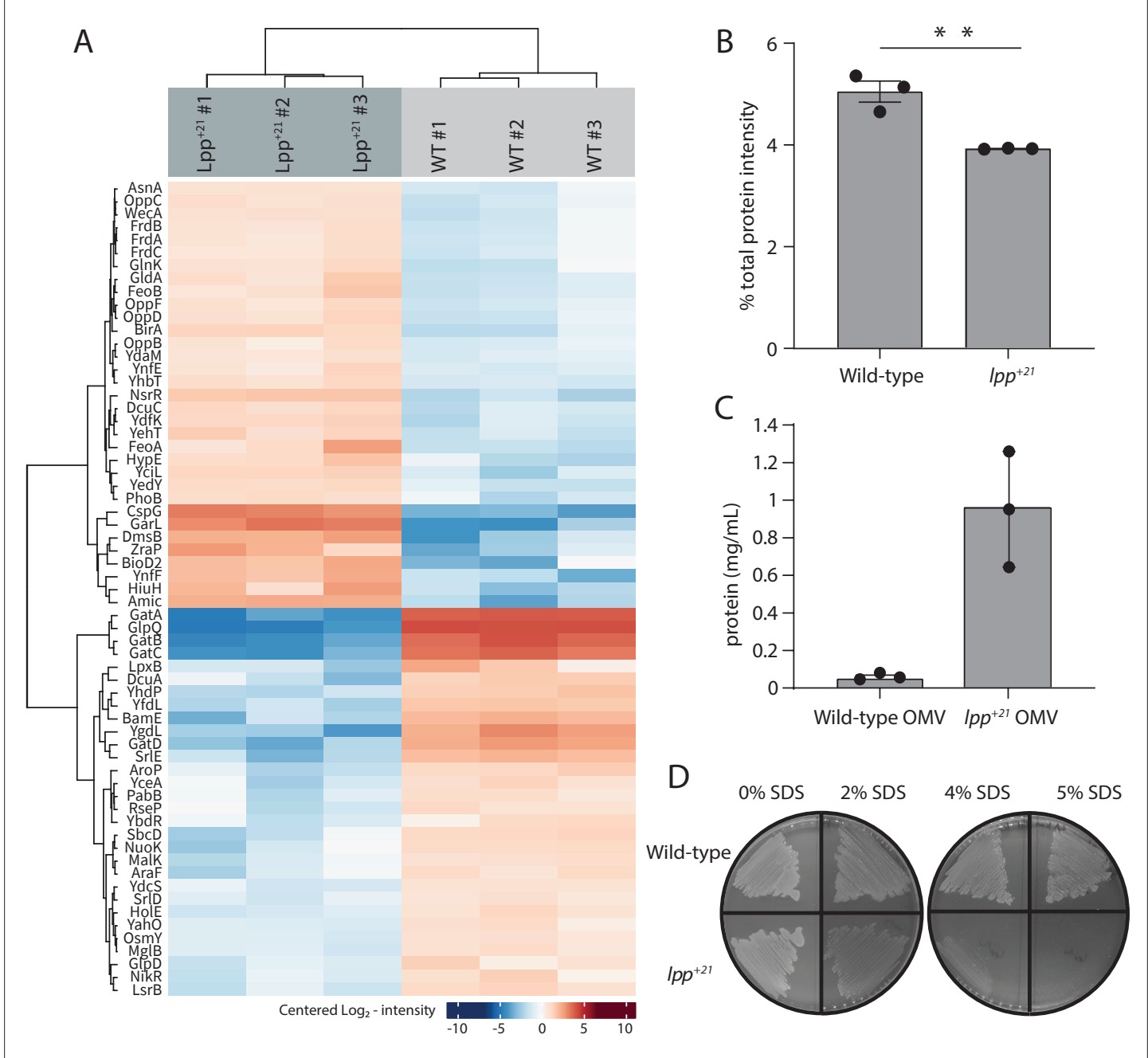

**Figure 2.** The Lpp+21 cells have altered outer membranes and increased blebbing. (**A**) Heat map of the significant proteomic differences observed between the wild-type and Lpp+21 mutant. Blue boxes indicate a relative reduction and red indicates a relative increase in protein level, centred on the average of the replicate samples. The grouping of the proteins is based on the similarity of the change in expression observed. (**B**) The Lpp+21 mutant has an overall reduction in the level of periplasmic proteins. (**C**) The Lpp+21 mutant has an increase in protein secreted via OMV blebbing. (**D**) SDS sensitivity profiles of the Lpp+21 mutant compared to the wild-type in increasing concentration of SDS in LB (solid) media. Representative data are shown from experiments performed in triplicate.

The online version of this article includes the following figure supplement(s) for figure 2:

**Figure supplement 1.** Quantitation of Lpp and Lpp+21 isoforms.

**Figure supplement 2.** Subcellular proteomics of Lpp+21 *E. coli*.

**Figure supplement 3.** Proteomics quality control report.

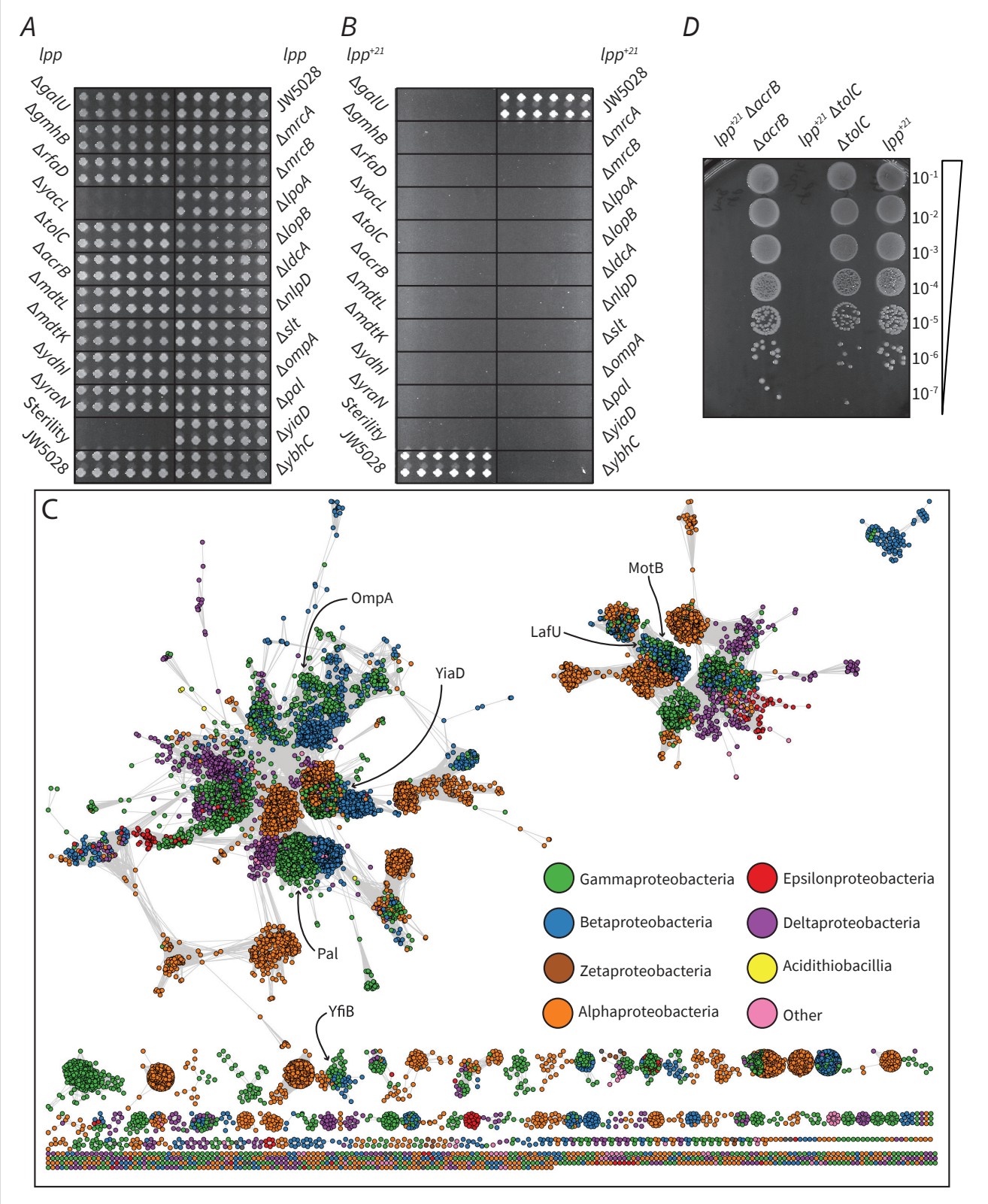

**Figure 3.** Factors that become essential to mediate OM-PG linkage in Lpp$^{+21}$ *E. coli*. (**A**) The growth phenotype in M9 minimal media (0.5 M sorbitol) of single gene knock outs that exhibit essentiality on Lpp$^{+21}$ background (*Table 1*). (**B**) The growth phenotype in M9 minimal media (0.5 M sorbitol) of double gene mutants. The mutants are results of Hfr Cavalli *lpp:lpp$^{+21}$* cat crossed with 22 Kan$^R$ recipients shown in panel A (Materials and methods). The double mutants are indicated and are arranged in four biological replicates (each having four technical replicates). (**C**) Sequence similarity network

*Figure 3 continued on next page*

*Figure 3 continued*

of domain (Pfam PF00691) containing proteins from across the Proteobacteria. Each circle represents a protein from a representative proteome (rp35) containing the PF00691 domain, connected by lines with a length imparted by their similarity score as defined by EFI - Enzyme Similarity Tool (*Gerlt et al., 2015*), with a cutoff of 30. Proteins are colored by their taxonomic class and the approximate location of the *E. coli* K12 six PF00691 proteins is indicated. (**D**) Synthetic lethal phenotype of the drug efflux mutants in the absence of selective antibiotics in M9 minimal media condition. Representative data are shown from experiments performed in biological triplicate.

The online version of this article includes the following figure supplement(s) for figure 3:

**Figure supplement 1.** A synthetic lethal screen to determine genes essential to Lpp+21 *E. coli*.

**Figure supplement 2.** Construction and characterization of the validation *yiaD* mutant.

**Figure supplement 3.** Synthetic lethality of major penicillin-binding proteins and their cognate lipoprotein activators with Lpp +21.

The only cytoplasmic factor identified in our screen, YraN is predicted to be a Holliday-junction resolvase related protein, and we therefore speculate that this mutant failed to resolve the merodiploid condition transient in the introduction of the $lpp^{+21}$ condition to the background strain, making the *yraN* mutant a technique-relevant artefact of the screen. This being the case, only functions performed in the periplasm were recovered as essential to viability for the Lpp$^{+21}$ strain.

Most of the components of the LPS biosynthetic machinery are essential genes in *E. coli* and are thus not represented in the library of non-essential genes. Several non-essential genes in the LPS biosynthetic pathway that are in the library, become essential to the Lpp$^{+21}$ strain (**Table 1**): the core LPS biosynthesis factors GalU, GmhB and RfaD were shown to be essential in the Lpp$^{+21}$ strain.

## An essential role for keeping the OM-PG distance

Several genes encoding proteins that could play roles in anchoring the PG within the cell envelope were identified as essential in the Lpp$^{+21}$ background. Independently, none of the major proteins bridging the OM and PG are essential for growth in *E. coli* (*Baba et al., 2006*) and all are therefore represented in the library. In a Lpp$^{+21}$ background the genes encoding the β-barrel protein OmpA and the lipoprotein Pal become essential (**Table 1**). PG-binding domain PF00691 is common to these proteins: appended to a beta-barrel in OmpA, but to a lipoyl anchor in Pal, and is also conserved in other proteins across diverse Gram-negative bacteria. In *E. coli*, there are four additional proteins

**Table 1.** Essential genes in the Lpp$^{+21}$ strain.

| Cellular process | Lpp$^{+21}$-essential genes | Function |
|---|---|---|
| LPS biosynthesis | galU | UDP-glucose metabolic process |
| | gmhB/yaeD | ADP-L-glycero-β-D-manno-heptose biosynthetic process |
| | rfaD/hldD/waaD | ADP-L-glycero-β-D-manno-heptose biosynthetic process |
| Peptidoglycan biosynthesis, turnover, and remodeling | lpoA, lpoB, | Regulators of PG synthases |
| | mrcA, mrcB | PG synthases |
| | ldcA | L,D-carboxypeptidase involved in PG recycling/turnover |
| | nlpD | Regulator of AmiC PG hydrolase |
| | slt | Lytic PG transglycosylase, degradation of uncrosslinked glycan strands |
| | acrB, mdtL*, mdtK*, ydhl* | IM components of drug efflux pumps. |
| | tolC | OM component of drug efflux pump |
| | ompA | β-barrel protein with PG binding domain |
| | pal | Lipoprotein with PG binding domain |
| PG-OM linkage | yiaD | Lipoprotein with PG binding domain |

*Putative IM drug efflux machinery, TolC independent with as yet unknown OM component (*Bay et al., 2017*).

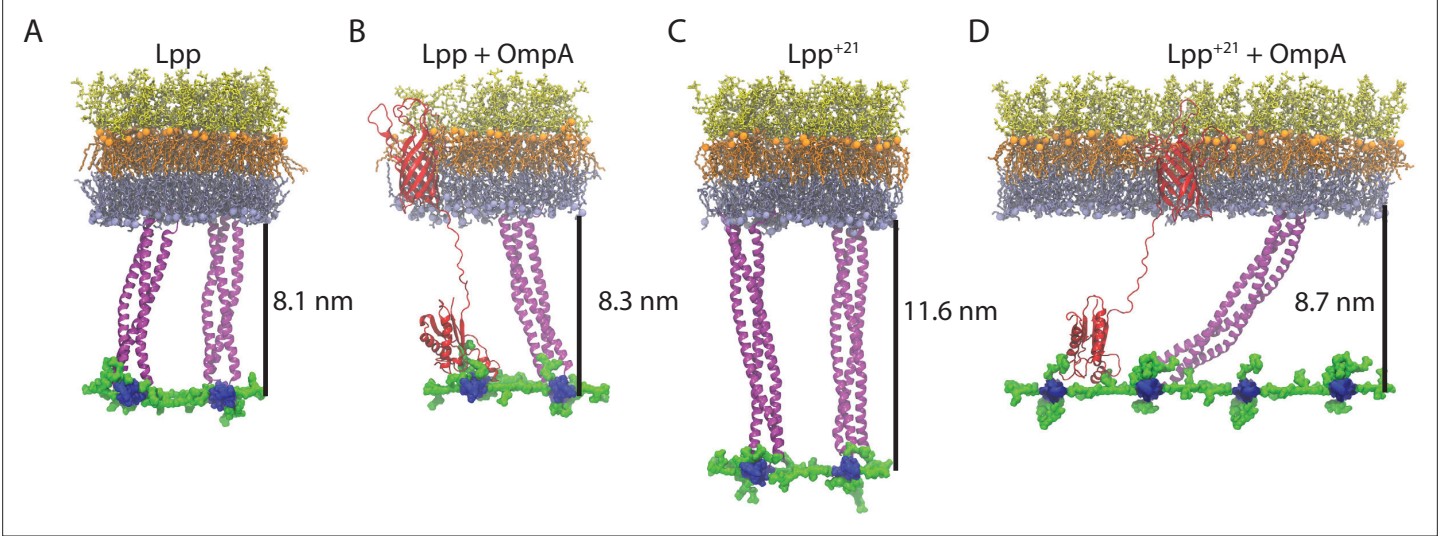

**Figure 4.** Final states of the OM-PG linkage from MD simulations. (**A–B**). A patch of OM with the LPS molecules depicted in orange (lipid A moiety) and yellow (core oligosaccharides), and the phospholipids in the inner leaflet of the OM depicted in gray. The PG layer (blue for glycans and green for peptide crosslinks) is attached to the OM via two trimers of Lpp (**A**), or a trimer of Lpp and the PG-binding domain of OmpA (red). The β-barrel anchor of OmpA is shown embedded in the OM. (**C–D**) Equivalent scenarios formed with Lpp$^{+21}$ trimers. The distances shown are calculated from the inner face of the OM to the centre of the PG layer and represent the average over the last 100 ns of a 200-ns simulation.

The online version of this article includes the following figure supplement(s) for figure 4:

**Figure supplement 1.** Molecular dynamics plots of Lpp tilt angles over time.

containing this PG-binding domain and these were mapped in a sequence similarity network analysis (*Figure 3C*). A protein of unknown function, YiaD, is present and it too is essential in a Lpp$^{+21}$ background (*Table 1*). We suggest, therefore, that this protein plays a substantive role in OM-PG linkage. The remaining three proteins: MotB, LafU and YfiB, are more divergent to the OmpA/Pal/YiaD cluster. Neither *motB*, *lafU* nor *yfiB* displayed a synthetic phenotype with Lpp$^{+21}$, and it has been suggested previously that *motB*, *lafU,* and *yfiB* are not expressed at detectable levels under laboratory conditions (*Li, 2013*).

Detecting genes encoding drug-efflux pumps as important for growth of the Lpp$^{+21}$ strain was initially surprising. Either the absence of the inner membrane proteins AcrB or the OM component TolC caused a reduction of growth in the Lpp$^{+21}$ genetic background (*Table 1*). When antibiotic selection was removed by plating the mutants on medium without chloramphenicol, the synthetic growth defects were observed in the absence of drug selection (*Figure 3D*), indicating that this synthetic phenotype is not the result of a decreased drug efflux activity. The trans-envelope AcrAB-TolC multidrug efflux pump has been shown to traverse through the PG and interact directly with PG at several defined sites (*Xu et al., 2012*; *Kim et al., 2008*; *Jo et al., 2019*; *Shi et al., 2019*; *Gumbart et al., 2021*), and as loss of the core AcrB and TolC components became essential, we suggest that this system could be acting as an additional OM-PG linkage that becomes essential in a Lpp$^{+21}$ background. Together with the observation that OmpA, Pal, YiaD, and TolC are also essential in the Lpp$^{+21}$ genetic background, these data suggest that functions that maintain local areas of closer contact between the OM and PG are essential for viability.

To compare the behaviors of Lpp and Lpp$^{+21}$, both with and without an OmpA monomer present, molecular dynamics (MD) simulations were run for 200 ns each (see Materials and methods for details). The Lpp lipoprotein from *E. coli* is a triple coiled-coil that is anchored to the inside face of the outer membrane by its N-terminal acyl group with a length equating to approximately 7.5 nm (*Shu et al., 2000*). The experiments were established to test the scenario for Lpp trimers or Lpp$^{+21}$ trimers in the absence or presence of an OmpA tether between the patch of OM and patch of PG (*Figure 4*). These MD simulations only represent a portion of the membrane, and as such we could not account for potential heterogeneity in the protein and LPS concentrations between a WT and Lpp$^{+21}$ cell. For three of the systems: Lpp only, Lpp with OmpA, and Lpp$^{+21}$ only, tilting was marginal, with tilt angles

of 76.9 ± 4.7°, 75.5 ± 4.7°, and 82.8 ± 2.9°, respectively (all numbers from the last 100 ns of the 200-ns trajectory). These angles are in agreement with previous simulations of Lpp alone (~80°) and Lpp with an OmpA monomer (~75°) PG (*Gumbart et al., 2021*; *Boags et al., 2019*; *Samsudin et al., 2017*). In initial simulations of Lpp$^{+21}$ with OmpA, the non-covalent connection between OmpA and PG was quickly disrupted as Lpp$^{+21}$ extended from its kinked state. Therefore, the simulation was repeated with an enforced OmpA-PG connection. Lpp$^{+21}$ was observed to both straighten and tilt within the first 100 ns; the tilt angle measured for the last 100 ns was 49.4 ± 2.3°.

The distance between the OM inner leaflet phosphorus atoms and the PG sugars was measured in each scenario. In the presence of Lpp, the distances with and without OmpA were similar at 8.3 ± 2.1 nm and 8.1 ± 1.2 nm, respectively. This was not true for the other scenarios where the distance for Lpp$^{+21}$ alone was 11.6 ± 1.3 nm, but for Lpp$^{+21}$ with OmpA, the distance was reduced significantly to 8.7 ± 2.7 nm. Thus, we observe that PG-binding proteins like OmpA can counteract the increased distance imposed by Lpp$^{+21}$, inducing it to tilt significantly in accommodation. We also compared our simulations to the distances that were measured by EM (centre of the OM to centre of the PG). In wild-type *E. coli* (i.e. Lpp+ OmpA), the centre-centre distance in the simulations is 10.7±2.2 nm (*Figure 4B*), similar to the 9.7–10.8 nm measured in intact cells (*Figure 1F*). The centre-centre distance measured in the simulation of Lpp$^{+21}$ tilted by the presence of OmpA (11.0±2.6 nm), fits the observed distances of 10.8–12.8 nm much better than the distance that would be created by a perpendicular Lpp$^{+21}$ (14.2±1.3 nm).

## Discussion

We observed that over a range of osmotic conditions, and in nutrient-rich or nutrient-poor media, growth rates of the Lpp$^{+21}$ strain of *E. coli* were equivalent to the isogenic wild-type *E. coli*, suggesting bacteria can adapt to the presence of the extended Lpp$^{+21}$. We observed a moderate increase to the OM-IM space similar to that previously reported (*Cohen et al., 2017*; *Asmar et al., 2017*; *Mathelié-Guinlet et al., 2020*) and reviewed (*Mathelié-Guinlet et al., 2020*; *Asmar and Collet, 2018*). Three adaptive features were expressed as phenotypes in the Lpp$^{+21}$ strain: (i) the steady-state level of the Lpp$^{+21}$ tether was reduced eightfold compared to the level of Lpp in the isogenic wild-type strain, and other tethers that enforce a wild-type distance: OmpA, Pal, YiaD, and TolC, became essential factors in the Lpp$^{+21}$ condition, (ii) structures that depend on a wild-type OM-IM distance, such as the LPS transport system, continued to function but key components of the system became essential for cell viability, and (iii) the PG network took on characteristics of dysregulated synthesis and all components of an otherwise redundant PG biosynthesis pathway become essential to viability. In response to Lpp elongation, we demonstrated a reduction in Lpp levels, as shown by the quantitative proteomic presented here, and the apparent tilting of the elongated Lpp as suggested by our molecular dynamics simulations. The reduction in the copy number of the elongated Lpp isoform observed in the quantitative proteomics was not due to a transcriptional downregulation of the *lpp$^{+21}$* gene, as RNAseq data demonstrated a moderate increase in the level of *lpp$^{+21}$* transcripts in the mutant (*Supplementary file 6*). Previous studies have not noted a marked reduction in Lpp$^{+21}$, but these studies also did not directly measure the levels of the protein (*Asmar et al., 2017*). One interpretation of the reduction of the level of Lpp$^{+21}$ observed here is that the Lpp$^{+21}$ isoform has a higher turnover in the periplasm, either via an inherently lower stability or driven by a targeted degradation due to its influences on the cell envelope structure. Taken together, this suggests a reinterpretation of previous conclusions drawn from experiments with elongated Lpp isoforms that assumed that Lpp$^{+21}$ was present at the same amount as Lpp, and that Lpp$^{+21}$ sits perpendicular to the membranes such that the 21 extra amino acids always stretch the periplasmic width by a constant amount (*Cohen et al., 2017*; *Asmar et al., 2017*; *Mathelié-Guinlet et al., 2020*).

### Tethers that enforce the distance constraint between the OM and PG layer became essential factors in the Lpp$^{+21}$ condition

The lipoyl-N-terminus of Lpp is integrated into the OM, and the C-terminus of a protomer of the Lpp trimer is covalently linked to the PG layer. Stereochemically, only one unit of a Lpp trimer can be covalently attached to the PG layer, and this stoichiometry has been observed experimentally (*Shu et al., 2000*; *Boags et al., 2019*; *Inouye et al., 1972*). This makes each Lpp trimer an important

bridge between the OM and PG layers of the cell wall, but it has not been clear whether the core role of Lpp is acting in compression or tension. In other words (*Miller and Salama, 2018*), is the role of Lpp to be a supportive brace to keep the OM away from the PG, or a binding anchor to bring the OM as closely as possible to the PG? In a wild-type *E. coli* scenario, molecular dynamics simulations show that the flexible linkers in PG-binding proteins like OmpA allow for adjustments in positioning the PG relative to the OM. In the presence of Lpp, the flexible linker of the PG binding domain of OmpA needs to extend further from the OM, as the distance between the OM and PG increases when Lpp sits perpendicular to the OM (*Boags et al., 2019*). In the absence of Lpp, OmpA determines the distance between the OM and PG (*Boags et al., 2019*; *Samsudin et al., 2017*) but is not essential for cell viability (*Babu et al., 2014*; *Babu et al., 2011*). Taken together with the data presented here from characterization of the Lpp$^{+21}$ phenotypes, we suggest that Lpp functions as a brace to keep the OM away from the PG layer.

The genetic screen showed that each one of the OM-located PG-binding factors Pal, YiaD, and OmpA are essential (as is TolC) for viability of the Lpp$^{+21}$ strain of *E. coli*, while previous high-throughput synthetic genetic screens demonstrate none of Pal, YiaD, OmpA nor TolC are essential or negatively affect the growth in an Lpp null background (*Babu et al., 2014*; *Babu et al., 2011*). The overlap in the distance constraints measured for cells expressing Lpp and Lpp$^{+21}$ suggests that much of the Lpp$^{+21}$ is in the highly tilted (49.4 ± 2.3°) form observed in the MD simulations. The enforced tilt and reduction in steady state levels of Lpp$^{+21}$ would severely impact its bracing force, with the measurable consequence that membrane integrity (SDS resistance) is diminished, OmpA and other tethers become essential to viability, and the OM is highly permissive to OMV formation. Furthermore, the discovery of YiaD in the genetic screen for essential factors is significant. Our proteomic assessment and previously published data (*Li et al., 2014*) demonstrated that the copy number of YiaD is much lower than that of either Pal or OmpA, making it not immediately clear how a minor component of the membrane could be exerting such an essential role. Previously, YiaD was suggested to be a factor regulating OMP biosynthesis by the BAM complex (*Tachikawa and Kato, 2011*). Structural analysis shows YiaD to be highly similar to the PG-binding domains of Pal and OmpA (*Ishida et al., 2014*). We suggest that the primary function of YiaD is to mediate OM-PG linkage, and that this indirectly impacts OMP biogenesis.

## Continued function of LPS transport in localized regions of the periplasm

The bridges needed to mediate LPS transport appear to be susceptible to disruption caused by Lpp$^{+21}$ in *E. coli*. LptA is subject to degradation if the Lpt complex is compromised (*Sperandeo et al., 2011*), with LptA-LptC and LptA-LptD interactions proposed as key quality control steps in the assembly of the Lpt complex (*Sperandeo et al., 2017*). That the Lpp$^{+21}$ cells have sufficient LPS in the OM to maintain membrane integrity is supported by our observations of the only minor increase in SDS sensitivity of the strain and observations by others of only minor changes in vancomycin sensitivity (*Mathelié-Guinlet et al., 2020*). These findings are consistent with observations through electron microscopy that regions of the periplasm in Lpp$^{+21}$ cells are maintained with OM-PG distances reflective of the wild-type condition, which would permit LPS transport to the OM.

## Components of the PG biosynthesis pathways become essential to cell viability

In *E. coli*, the weave of the PG-layer is maintained in a uniform, open state through quality control mechanisms that depend on the regulators in the OM (LpoA and LpoB) being able to permeate it to contact the synthetases (PBP1a and PBP1b) in the IM. It has been hypothesized in this way that the OM lipoproteins may serve as a molecular ruler to modulate PG thickness, maintaining a single layer of PG equidistant from the OM layer under normal conditions (*Typas et al., 2011*). The PG synthases encoded by *mrcA* and *mrcB* are redundant in the sense that *mrcA* mutant strains and *mrcB* mutant strains are each viable (*Cho et al., 2016*; *Mueller et al., 2019*; *Suzuki et al., 1978*). However, in the Lpp$^{+21}$ strain both *mrcA* and *mrcB* (as well as their OM lipoprotein partners) are essential for viability, indicative of a compromised capability to build the PG-layer. Mutations designed to impact on these interactions lead to transient deposition of 'high-density PG' and 'multi-layered PG' through dysregulation of the synthetases (*Typas et al., 2011*). The broader and more diffuse morphology of

the PG-layer observed by electron microscopy in the Lpp$^{+21}$ strain could suggest a similar phenotype associated with transient or local impacts on the OM-PG distance (*Typas et al., 2011*). It is worth noting Δ*lpp* Δ*mrcB* mutants have previously shown a moderate growth defect (*Chen et al., 2014*) and that this defect could be alleviated with the introduction of the *lpp*$^{+21}$ gene (*Asmar et al., 2017*), whereas in the present study we observed a synthetic lethal phenotype in the Δ*mrcB lpp*$^{+21}$ mutants. This discrepancy is potentially due differences in growth conditions used (LB vs hyperosmotic minimal media) imparting additional periplasmic stress or differences in the strain background (DH300 vs BW25113). Indeed, the Δ*mrcB lpp*$^{+21}$, Δ*mrcA lpp*$^{+21}$, Δ*lpoA lpp*$^{+21}$, and Δ*lopB lpp*$^{+21}$ mutants in the current study could grow on LB media but not on minimal media (*Figure 4—figure supplement 1*).

An essential requirement was also placed on PG-layer remodeling, whereby the PG-binding factor NlpD was found to be essential in Lpp$^{+21}$ cells: its function is in modulating the activity of the amidase AmiC (*Stohl et al., 2016*; *Yang et al., 2018*) to remodel PG strands, and AmiC was observed at increased steady-state levels in the Lpp$^{+21}$ strain. Taken together with the increase in oligopeptide transporter subunits (OppB, OppC, OppD, and OppF) in the Lpp$^{+21}$ strain to recycle PG precursors across the IM, our results suggest a clearance of the malformed PG caused by dysregulation of the PG synthases is a crucial adaptation in the Lpp$^{+21}$ strain.

### Stiffness and connection of OM-PG

The concept of bacterial cell stiffness has emerged as a means to understand the physical parameters that define how readily bacteria can respond to major environmental changes (*Rojas et al., 2017*; *Rojas and Huang, 2018*; *Hwang et al., 2018*). Measurements by AFM have revealed a characteristic stiffness in Gram-negative bacterial cells that is contributed by load-bearing outer membrane and its attachment to the underlying PG layer (*Rojas et al., 2018*). In *E. coli*, mutants lacking Lpp, Pal, or OmpA are softer than wild-type cells (*Rojas et al., 2018*), and cells expressing the Lpp$^{+21}$ isoform are also softer than wild-type cells (*Mathelié-Guinlet et al., 2020*). Taken together with our observation that the Lpp$^{+21}$ isoform display a broader and more diffuse PG layer morphology in the subtomogram averages, this further supports the proposition that the OM is a major contributor to cell stiffness (*Rojas and Huang, 2018*; *Rojas et al., 2018*).

Although Lpp length alone does not appear to dictate periplasm width, indeed the *Pseudomonas aeruginosa* Lpp protein is only four amino acids longer than that of *E. coli* but its periplasm has been measured as ~3 nm larger under identical conditions (*Matias et al., 2003*), the length of Lpp seen in *E. coli* is conserved in a range of bacterial lineages. The species from the genus *Geobacter* naturally express longer Lpp proteins of 99 or more residues, equivalent to Lpp$^{+21}$ in length. *Geobacter* species have a complicated periplasm housing 'electron conduits' formed of transmembrane and periplasmic redox proteins, in order to transfer electrons to the external surface of the bacterial cell (*Jiménez Otero et al., 2018*; *Santos et al., 2015*). There have been no direct measurements of the width the periplasm of *Geobacter*, but it would be interesting to see if these naturally longer Lpp forms correlate with an enlarged periplasm. That species of *Geobacter* serve as an exception to what is otherwise a strict rule about the length of Lpp, and thus the structurally enforced distance constraint between the OM and PG layer, raises interesting questions about whether OM softness, OMV production or increased OM permeability might assist the unusual biological functions of the OM and periplasm in bacteria other than *E. coli*.

## Materials and methods
### Bacterial strains and growth conditions

*E. coli* BW25113 was the parental strain of all the recipient strains used in this study. JW5028, a derivative of BW25113, containing a kanamycin resistance marker in place of a pseudogene (*Gagarinova et al., 2012*), was used as the wild-type for this study. Strains were grown in LB broth or M9 minimal media broth supplemented with 1 mM MgSO$_4$, 0.1 mM CaCl$_2$, 1.12 mM thiamine and 0.2% (w/v) glucose for a defined minimal media. To osmotically stabilize the growth medium, sorbitol was supplemented to a final concentration of 0.5 M. For overnight cultures, strains were grown in LB broth overnight at 37°C under continuous agitation. Subculturing was done by diluting the saturated culture 1:100 using new media. The cells were then grown to mid-exponential growth phase (OD$_{600}$ = 0.5–0.6) at 37°C under shaking. Culture media was supplemented with antibiotics for plasmid selection and

maintenance or selection of mutants at the following concentrations: 100 µg/ml ampicillin, 30 µg/ml kanamycin, 34 µg/ml chloramphenicol. Fifteen g/l agar was added to media before autoclaving when solid media was required (*Figure 3—figure supplement 1*).

## Construction of Lpp$^{+21}$ mutant

The endogenous *lpp* gene was replaced with the extended *lpp*$^{+21}$ gene previously described (*Cohen et al., 2017*; *Asmar et al., 2017*) with minor modifications (*Figure 1—figure supplement 1*). First the replacement was done in the donor *E. coli* Hfr Cavalli cells using the $\lambda$ –red recombination system (*Datsenko and Wanner, 2000*). A gene block was sourced (Integrated DNA Technologies) containing extra 21 amino acid residues (three heptad repeats) inserted between codon 42 and 43 of *E. coli* Lpp. The gene block also contained 50 bp DNA flanking 5' and 3' ends of the *lpp*$^{+21}$ gene. On the 5' end, the extension was homologous to DNA sequence upstream of *lpp*, while on the 3' end, the extension was homologous to the *cat* gene. The gene block was combined with the *cat* gene by Gibson assembly, and the resulting PCR fragment was used to replace *lpp*. The new Lpp$^{+21}$ strain was selected by plating on medium containing chloramphenicol. Chromosomal *lpp*$^{+21}$ was moved into *E. coli* BW25113 background by mating with kanamycin-resistant Keio collection strain JW5028, described above, to generate a double mutant (*Baba et al., 2006*; *Gagarinova et al., 2012*; *Figure 3—figure supplement 1*). The mutation was verified by PCR described (*Figure 1—figure supplement 1*) and sequencing.

## SDS-PAGE and immunoblotting

Cells grown in M9 minimal media (0.5 M sorbitol) and normalized by OD$_{600nm}$, lysed in Laemmli SDS-loading dye and separated in 15% acrylamide gels and transferred to 0.45 µm hydrophobic Immobilon-P PVDF membrane (Merck Millipore). Immunoblotting was as described previously (*Baba et al., 2006*). Rabbit primary antibodies; α-Lpp antibody (kindly provided by T. Silhavy) and α-OmpA were diluted 1:400,000 and 1: 30,000, respectively in 5% skim milk, TBST. The membranes were incubated with goat, α-rabbit IgG, HRP-conjugated secondary antibody (Sigma; 1: 20,000 in 5% skim milk, TBST), and washed with TBST. Detection was by enhanced chemiluminescence with ECL prime western blotting detection reagent (GE Healthcare Life Sciences), visualized using Super RX-N film (Fujifilm).

## Outer membrane vesicle purification and quantification

Overnight cultured cells, grown in LB without antibiotics, were washed twice in 1 x M9 salts then subcultured in 500 ml M9 minimal media supplemented with 0.5 M sorbitol (1:1000 dilution). The strains were grown to late logarithmic phase without antibiotics, OD$_{600}$ ≈ 0.9 and spun down to collect culture supernatant. Collected culture supernatant were then processed for OMVs isolation and purification using differential ultracentrifugation technique as discussed previously (*Deo et al., 2018*). OMVs were washed twice in PBS to remove sorbitol then quantified using a bicinchoninic acid assay kit (Thermo Scientific CST#23225).

## SDS sensitivity

Streaks were made on LB (solid) media with 0% SDS – 5% SDS. After 16 hr of incubation at 37°C, the minimum SDS concentration inhibiting growth was obtained by analyzing growth on each concentration. The streaks were done in duplicate and repeated three times.

## Proteomics

Saturated overnight cultures were washed twice in 1 x M9 salts and diluted 1:100 in 10 ml M9 minimal media supplemented with 0.5 M sorbitol. Cultures were further grown to logarithmic phase, collected by centrifugation and washed using PBS buffer. The cell pellet was homogenised in 4% SDS, 100 mM Tris, pH 8.1 and boiled at 95 °C for 10 min. The lysate was then sonicated with a Bioruptor Pico (Diagenode) and protein concentration was determined with Bicinchoninic Acid assay (BCA, Thermo Fisher). SDS was removed with chloroform/methanol, the protein was digested by trypsin overnight and the digested peptides were purified with ZipTips (Agilent). Using a Dionex UltiMate 3000 RSLC-nano system equipped with a Dionex UltiMate 3000 RS autosampler, an Acclaim PepMap RSLC analytical column (75 µm x 50 cm, nanoViper, C18, 2 µm, 100 Å; Thermo Scientific) and an Acclaim PepMap 100 trap column (100 µm x 2 cm, nanoViper, C18, 5 µm, 100 Å; Thermo Scientific), the tryptic peptides

were separated by increasing concentrations of 80% ACN / 0.1% formic acid at a flow of 250 nl/min for 120 min and analyzed with a QExactive Plus mass spectrometer (Thermo Scientific). The instrument was operated in the data dependent acquisition mode to automatically switch between full scan MS and MS/MS acquisition. Each survey full scan (m/z 375–1575) was acquired in the Orbitrap with 60,000 resolution (at m/z 200) after accumulation of ions to a $3 \times 10^6$ target value with maximum injection time of 54ms. Dynamic exclusion was set to 30 s. The 20 most intense multiply charged ions ($z \geq 2$) were sequentially isolated and fragmented in the collision cell by higher-energy collisional dissociation (HCD) with a fixed injection time of 54ms, 15,000 resolution and automatic gain control (AGC) target of $2 \times 10^5$.

The raw data files were analyzed using MaxQuant software suite v1.6.5.0 (*Cox and Mann, 2008*) against Andromeda search engine (*Cox et al., 2011*) for protein identification and to obtain their respective label-free quantification (LFQ) values using in-house standard parameters. The proteomics data was analyzed using LFQ-Analyst (*Shah et al., 2020*) and the analysis of the data quality analysis is presented in *Figure 2—figure supplement 3*. Due to the 21 amino acid insertion in the Lpp$^{+21}$ isoform, the relative levels of Lpp in the mutant had to be assessed manually. Only the unique peptide (IDQLSSDVQTLNAK) shared between the two isoforms was used to quantify the levels of Lpp and Lpp$^{+21}$ in the wild-type and mutant strain, respectively (*Figure 2—figure supplement 2*).

To estimate the total relative amount of proteins from the various subcellular compartments, the raw intensities from peptides identified from proteins from different subcellular locations were summed and divided by the total summed intensity from all peptides. Subcellular locations annotations were applied from the STEPdb 2.0 (*Loos et al., 2019*) where proteins designated F1, A, R, and N were classified as cytoplasmic; B was designated inner membrane; H, X, and F4 were designated outer membrane / extracellular; and I, G F2, F3, and E were designated as periplasmic.

## Sequence similarity network analysis

Proteobacterial proteins containing the Pfam domain (PF00691) were extracted from the Representative Proteome 35% co-membership rpg-35 group (*Chen et al., 2011*) and a sequence similarity network was generated with the EFI Enzyme Similarity Tool (*Gerlt et al., 2015*). This network was visualized with Cytoscape (*Shannon et al., 2003*) with a similarity score cutoff of 30. Each protein is represented by a colored circle node and each similarity match above the similarity score cutoff is represented by an edge between nodes with the length determined by the similarity score.

## Lpp length distribution across bacterial species

To determine the amino acid length distribution of Lpp in Gammaproteobacteria (*Supplementary file 4*), amino acid sequences were sourced from the InterPRO database (version 81.0) (*Mitchell et al., 2019*) using the Interpro Family tag - Murein-lipoprotein (IPR016367). Filtered Lpp sequences were then concatenated into representative nodes (at least >90% sequence similarity) using the online available amino acid Initiative-Enzyme Similarity Tool (EFI-EST) (*Gerlt et al., 2015*).

## Synthetic genetic interaction array

The Lpp$^{+21}$ isoform was transferred to each of the Keio collection clones by conjugation as described (*Figure 3—figure supplement 1*). First, the Hfr chloramphenicol resistant Lpp$^{+21}$ strain was arrayed in 384-colony density on LB agar containing chloramphenicol using the Singer rotor HAD (Singer Instruments, United Kingdom). Similarly, the Keio collection arrayed in 384-colony density was pinned on LB agar plates containing kanamycin and incubated overnight at 37°C. Using the Singer rotor HDA, the Hfr Lpp$^{+21}$ strain and the Keio collection clones from the 384-colony density were then co-pinned onto LB agar plates and incubated at 37°C for 16 hr. Following conjugation, the colonies were transferred to LB agar with kanamycin (selection 1) at the same colony density and incubated at 37°C for 16 hr. To select for double mutants (selection 2), colonies from the intermediate selection were pinned on LB agar with both kanamycin and chloramphenicol and incubated at 37°C for 14 hr. For assessment of synthetic genetic interaction in nutrient-limited media, the double mutants generated were replica pinned in M9 minimal media at the same density and incubated at 37°C for 25–30 hr. Images were acquired using Phenobooth (Singer Instruments, United Kingdom) for analysis. Images were manually screened to cross-reference recipient plate images to the final double antibiotic selection plates images. Candidate synthetic lethal or growth-compromised mutants were then subjected to another

round of screening in the same conditions as previously identified (mini-screen) for validation. Four biological replicates were included that were further arrayed in four technical replicates. Mutants were confirmed by PCR. Where further phenotypic screening of mutants was conducted, independent isogenic knock-out mutants were generated using the PCR and $\lambda$-red based homologous recombination method (*Datsenko and Wanner, 2000*).

Since the Keio collection *yiaD* mutant has been identified as containing a potential duplication event (*Yamamoto et al., 2009*), the candidate *yiaD* synthetic lethal interaction was confirmed through independently constructing a *yiaD* mutant in the BW25113 strain background (*Figure 3—figure supplement 2*). The *lpp*[+21] variant was subsequently generated in this mutant as described above. As with the Keio *yiaD* mutant, this strain demonstrated synthetic lethality on M9 media.

Two colony PCR reactions (*Datsenko and Wanner, 2000*; *Yamamoto et al., 2009*; *Figure 3—figure supplement 1*) confirmed the identity of all candidate double mutants, using a set of primers flanking the *lpp* gene, and a set of primers flanking the kanamycin gene (*Supplementary file 5*).

## Preparation of electron cryo-microscopy samples, data collection, and analysis

Strains were grown aerobically in M9 minimal media (0.5 M sorbitol) until an $OD_{600}$ of 0.6 was reached. Cells were collected by spinning at 6000xg for 5 min and resuspended to an $OD_{600}$ of $\approx$ 12. Cryo-EM, data collection, and analysis were performed similarly to previous studies (*Cohen et al., 2017*; *Asmar et al., 2017*), except using 3-D subtomogram averages derived from whole-cell cryotomograms instead of projection images so as to discern peptidoglycan. Tilt series of WT and Lpp[+21] strains was acquired on an FEI Krios operating at 300 keV with a Gatan K2 direct detector and energy filter with a 20 eV slit with a tilt range of±60° using 3° increments and reconstructed using IMOD. Subtomograms were picked manually using 3DMOD along the length of all non-polar periplasm and averaged using PEET.

## Generation of simulation systems

Initially, two systems were generated: the OM and PG, as previously detailed (*Hwang et al., 2018*), with two copies of wild-type Lpp and with two copies of Lpp[+21]. For wild-type Lpp, we used the homo trimer from PDB 1EQ7 (*Shu et al., 2000*). For Lpp[+21], a monomer was first built using I-TASSER (*Yang and Zhang, 2015*). Next, the trimer of Lpp[+21] was built using the wild-type Lpp trimer as a template, further optimized using Targeted Molecular Dynamics (TMD) for one ns. For both Lpp and Lpp[+21], the proteins were anchored in the OM via N-terminal acylation while the C-terminus of one copy from each trimer was covalently linked to the PG. The systems generated were prepared for equilibration using the following steps for one ns each: (1) minimization for 10,000 steps, (2) melting of lipid tails, (3) restraining only the PG and the protein, and (4) restraining the PG and the protein backbone. Both systems were equilibrated for 200 ns.

For each of the two systems (Lpp and Lpp[+21]), a new system was constructed with one Lpp trimer removed and OmpA inserted into the OM. The full-length OmpA structure was taken from *Ortiz-Suarez et al., 2016*. The periplasmic domain region of the OmpA (clamp, hereafter) was lowered to the PG by shortening the distance between the PG and the clamp for 20 ns. After the clamp was lowered, it was clenched to the nearest DAP residue by using two different distance collective variables (*Fiorin et al., 2013*) between the centre of mass of residue 242 or 256 from OmpA and that of a nearby PG DAP residue to maintain the connection for 110 ns. In the case of Lpp[+21], PG was first pulled toward the OM to match the distance between them in wild-type Lpp, after which the clamp of OmpA was lowered and clenched for 110 ns. The wild-type Lpp/OmpA system was equilibrated for 200 ns; OmpA stayed bound to the PG without colvars. The Lpp[+21]/OmpA system was also equilibrated for 200 ns, but colvars were needed to maintain the OmpA-PG interaction.

## Molecular dynamics (MD)

All-atom molecular dynamics simulations were performed using NAMD 2.11 (*Phillips et al., 2005*) and the CHARMM36m (*Huang et al., 2017*) and CHARMM36 (*Klauda et al., 2010*) force-field parameters for proteins and lipids, respectively, with the TIP3P-CHARMM water model (*Jorgensen et al., 1983*). Unless otherwise stated, all MD simulations were performed under a periodic boundary condition with a cut-off at 12 Å for short-range electrostatic and Lennard-Jones interactions with a force-based

switching function starting at 10 Å. For long-range electrostatic calculations, the particle-mesh Ewald method (*Darden et al., 1993*) with a grid spacing of at most 1 Å was used for long-range electrostatic calculations. Bonds between a heavy atom and a hydrogen atom were maintained to be rigid, while all other bonds remain flexible. Unless otherwise stated, each system was equilibrated under an isothermal-isobaric ensemble (NPT) at 310 K and 1 bar, with a timestep of 4 fs after hydrogen mass repartitioning (*Balusek et al., 2019*). A Langevin thermostat with a damping coefficient of 1 ps$^{-1}$ was used for temperature control and a Langevin piston was used for pressure control. VMD was used for all visualization and analysis (*Humphrey et al., 1996*).

## Acknowledgements

We are grateful to Rebecca Bamert and Jonathan Wilksch for critical comments on the manuscript. The authors thank Andrea Nans and Peter Rosenthal at the Francis Crick Institute for electron microscopy services, and support staff at the Monash Proteomic & Metabolomic Facility. We acknowledge research support from the Australian Research Council (FL130100038 to TL and IDH). EM was recipient of a Monash Research Scholarship. JCG acknowledges support from the US National Institutes of Health (R01-GM123169 and R01-AI052293). Computational resources were provided through XSEDE (TG-MCB130173), which is supported by the US National Science Foundation (NSF; ACI-1548562). This work also used the Hive cluster, which is supported by the NSF (1828187) and is managed by PACE at the Georgia Institute of Technology.

## Additional information

### Funding

| Funder | Grant reference number | Author |
|---|---|---|
| Australian Research Council | FL130100038 | Trevor Lithgow Iain D Hay |
| National Institutes of Health | R01-GM123169 | JC Gumbart |
| National Institutes of Health | R01-AI052293 | JC Gumbart |
| National Science Foundation | | JC Gumbart |
| Marsden Fund | UOA1907 | Iain D Hay |

The funders had no role in study design, data collection and interpretation, or the decision to submit the work for publication.

### Author contributions

Eric Mandela, Conceptualization, Data curation, Formal analysis, Investigation, Methodology, Visualization, Writing - original draft, Writing – review and editing; Christopher J Stubenrauch, Conceptualization, Data curation, Formal analysis, Investigation, Methodology, Funding acquisition, Software, Supervision, Visualization, Writing - original draft, Writing – review and editing; David Ryoo, Eli J Cohen, Formal analysis, Investigation, Methodology, Software, Visualization; Hyea Hwang, Von L Torres, Formal analysis, Investigation, Methodology, Visualization; Pankaj Deo, Formal analysis, Investigation, Methodology; Chaille T Webb, Data curation, Formal analysis, Software, Supervision; Cheng Huang, Formal analysis, Investigation, Methodology, Supervision, Visualization; Ralf B Schittenhelm, Data curation, Formal analysis, Investigation, Methodology, Software, Supervision, Visualization; Morgan Beeby, Formal analysis, Investigation, Methodology, Software, Visualization, Writing – review and editing; JC Gumbart, Conceptualization, Data curation, Formal analysis, Investigation, Methodology, Software, Visualization, Writing – review and editing; Trevor Lithgow, Conceptualization, Data curation, Visualization, Investigation, Funding acquisition, Project administration, Software, Supervision, Writing - original draft, Writing – review and editing; Iain D Hay, Conceptualization, Data curation, Formal analysis, Investigation, Methodology, Funding

acquisition, Project administration, Software, Software, Supervision, Visualization, Writing - original draft, Writing – review and editing

### Author ORCIDs
Christopher J Stubenrauch http://orcid.org/0000-0003-4388-3184
Von L Torres http://orcid.org/0000-0003-3387-1112
Pankaj Deo http://orcid.org/0000-0002-6947-5317
Morgan Beeby http://orcid.org/0000-0001-6413-9835
JC Gumbart http://orcid.org/0000-0002-1510-7842
Iain D Hay http://orcid.org/0000-0001-8797-6038

### Decision letter and Author response
Decision letter https://doi.org/10.7554/eLife.73516.sa1
Author response https://doi.org/10.7554/eLife.73516.sa2

---

## Additional files

### Supplementary files
• Supplementary file 1. Proteomic results.

• Supplementary file 2. Substantive changes in steady-state protein levels in cell envelope of Lpp$^{+21}$.

• Supplementary file 3. Proteins used in the generation of the sequence similarity network.

• Supplementary file 4. Representative Lpp protein information.

• Supplementary file 5. Bacterial strains and primers used in the study.

• Supplementary file 6. Genes up-regulated in Lpp +21 strain.

• Transparent reporting form

• Source data 1. Raw gel image files used for the generation of Figure 1, Figure S1, Figure S3, and Figure S5.

### Data availability
All data generated from this study is supplied in the relevant supplemental files.

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
