## [Editor Report]

In this study, Mandela et al. investigate the response of cells to a lengthened version of the periplasmic protein Lpp (Lpp^+21^). An abundant protein present in high abundance in cells, Lpp tethers the outer membrane to the peptidoglycan layer and is implicated in maintaining the distance between the two. Combining genetics, proteomics and simulations, the authors determine that lengthening Lpp does not change the spatial organization of the periplasm due to apparently compensatory effects. Together, these data highlight the importance of periplasmic organization to cell viability and the resilience of the systems that maintain it.

---

## [Decision Letter]

**Decision letter after peer review:**

[Editors’ note: the authors submitted for reconsideration following the decision after peer review. What follows is the decision letter after the first round of review.]

Thank you for submitting your work entitled "Periplasm homeostatic regulation maintains spatial constraints essential for cell envelope processes and cell viability" for consideration by *eLife*. Your article has been reviewed by 3 peer reviewers, and the evaluation has been overseen by a Reviewing Editor and a Senior Editor. The reviewers have opted to remain anonymous.

We are sorry to say that, after consultation with the reviewers, we have decided that your work will not be considered further for publication by *eLife*.

All three reviewers appreciated the genetic approach to understand how changes in the extended Lpp+21 variant modulates periplasmic space. At the same time, however, there was consensus that many of the results were preliminary and mechanistic insight was generally missing. We hope you will find the detailed comments provided by all three reviewers helpful.

*Reviewer #1:*

The outer-membrane anchored lipoprotein Lpp is the most abundant protein in *E. coli* and about one third of the Lpp molecules are covalently attached to peptidoglycan, which is important to maintain the outer membrane impermeable to SDS (anionic detergent). Lpp forms a homotrimer made of α-helical monomers, hence, the abundant connections between the peptidoglycan and the outer membrane via Lpp are thought to maintain a constant distance between both envelope layers, which is likely important for trans-envelope processes. In two previous papers, an extended version of Lpp, Lpp+21, was used to study flagellar assembly and signal transduction through an artificially widened periplasm but it remained unclear what other processes are impaired in the Lpp+21 expressing cells. Mandela and co-workers provide now a comprehensive characterization of the strain. Different from previous papers, they conclude that lengthening of Lpp only slightly affects the widths of the periplasm under hyperosmotic condition, but there is a more pronounced effect on the mechanical properties of the outer membrane. The expression of Lpp+21 induces several stress responses and renders specific components of the peptidoglycan remodelling pathway essential.

I find this an interesting and important study and would like to make the following specific comments:

Main points:

1. I am concerned that some of the phenotypes of the Lpp+21 expressing strain are caused by the ~8-fold reduced copy number, compared to the copy number of native Lpp in wild-type cells. With 8-fold reduced amount of Lpp, other outer membrane-peptidoglycan connecting proteins are likely more abundant than Lpp+21 and capable of maintaining a normal distance between both layers, explaining why the authors did not observe a more increased width in the Lpp+21 strain. It would be helpful to try to compare strains with similar amounts of Lpp+21 or Lpp and find out which phenotypes, proteome changes and changes in essentiality of genes are caused by the reduced amount of Lpp+21, and which are caused by the extension of the protein.

2. Line 113 and Figure 1F, numbers of periplasmic widths. The numbers measured by the authors are not numbers of periplasmic widths, as they include also half of the cytoplasmic membrane and half of the outer membrane (measurements were take from the middle of each membranes). Hence, these numbers are different to published data of periplasmic widths (for example, from the Beveridge lab), which could confuse the readers. Hence, I suggest to explain the difference to published numbers in the text and refrain from using the term 'periplasmic widths' for their measurements.

3. Line 115, PG morphology. It is hard to see whether the PG morphology is changed (Figure 1F) because the images of the wt are generally darker than those of the mutant. Can the authors apply some sort of image analysis to support their claims that there is a difference? However, it is also not clear what insights about the PG layer one can possibly obtain from transmission EM images and the discussion about possible effects is very speculative (lines 300-302).

4. Figure 1F. It might be interesting to compare the Lpp+21 strain to a mutant lacking Lpp?

5. Figure 4C/D, model. Are they sure that the attachment of Lpp+21 to peptidoglycan is as efficient as the attachment of wt Lpp? If many Lpp+21 trimers were not attached to peptidoglycan, this would explain the almost normal distance between outer membrane and peptidoglycan, the strong vesicle production (Figure 2C) and the higher SDS sensitivity (Figure 2D). Have the authors considered simulations with Lpp+21 that is not attached to peptidoglycan, would it penetrate the layer?

*Reviewer #2:*

The outer membrane lipoprotein Lpp provides the only covalent link known to exist between the outer membrane and the peptidoglycan cell wall in *E. coli*. It has been previously shown that elongating the coiled-coil structure of Lpp increases the distance between the outer membrane and the cell wall, increasing the overall width of the periplasm. Mandela et al. investigate the effects of increasing the length of Lpp by 21 (Lpp+21) residues using proteomics, a genetic screen for factors that become essential when Lpp is elongated, and molecular dynamics. Based on their results, the authors propose that factors that maintain the distance constraint between the outer membrane and the cell wall become essential. However, they do not provide clear insight into why other factors were identified in their experiments.

The authors set out to investigate an interesting question and present a significant amount of data. However, there are important concerns about the study.

1) Important phenotypes reported in the study are significantly different from data previously reported by others. Although the authors acknowledge some (not all) of the discrepancies in the Discussion, they do not address the differences. These differences relate to the distribution, uniformity, and overall values of the distance between the outer membrane and the peptidoglycan layer, the impact that adding 21 residues to Lpp has on Lpp protein levels, and the reported synthetic lethality with mrcB. It would be necessary to address why these differences exist to validate this study.

2) The authors provide a list of genes and proteins from the experiments that they conducted using proteomics and genetics. However, they do not further explore the relationship or mechanism linking those factors to Lpp+21. As a result, the work is limited to reporting correlations and speculative links between factors and lengthening Lpp. None of the proposed connections between Lpp+21 and the identified factors are investigated. As a result, it is unclear whether the effect results from a direct or indirect link to the presence of Lpp+21, or even whether they are the result of increasing the distance from the OM to the cell wall, or simply because the levels of Lpp+21 are significantly decreased in comparison to those of wild-type Lpp in the parent strain.

3) The authors conducted a synthetic lethal screen with non-essential genes and lpp+21. The screen relied on conjugation to construct double mutants. However, the authors have not demonstrated that the recipient single mutant strains are not defective in conjugation, which would lead to the wrong conclusion.

4) The conclusions that OmpA becomes essential in the presence of lpp+21 because it anchors the outer membrane to the cell wall is plausible, but requires further testing, since that is not the only role that OmpA plays.

5) The authors incorrectly conclude that "the" (i.e. "all") non-essential genes involved in LPS synthesis become essential in the lpp+21 strain in the Results and Discussion sections. There are several non-essential genes involved in LPS synthesis that were not identified in this work. Moreover, the authors do not explain why changing LPS structure lead to synthetic lethality. It is also unclear why galU, which is required for making the outer core, would be essential but genes encoding for enzymes required for the synthesis of the middle core are not when they function upstream in the LPS biosynthesis pathway.

6) The authors propose that the deletion of some factors leads to synthetic lethality because they are involved in making complexes that link the inner and outer membranes. The authors suggest that the reason why pumps were identified is because they form a complex linking these membranes. However, AcrA forms a pump with AcrB and TolC but acrB was not identified in the screen. It would be good to know what is the evidence that MdtK, MdtL and YdHI form pumps that connect the two membranes.

Recommendations for the authors:

1) Some of the crucial data presented here are different from data previously reported by others. The authors acknowledge this in the Discussion but provide no explanation, which is very problematic. The most salient differences relate to the phenotypes caused by the presence of the lpp+21 allele. The distribution of the width of the periplasm in both the wild-type and the lpp+21 strains is larger than previously reported (e. g. https://pubmed.ncbi.nlm.nih.gov/29257832/. ). In addition, Asmar et al. (PMID: 29257832) reported that levels of the Lpp+21 protein are similar to those of wild-type Lpp. Here, the authors report that they are significantly lower. Given that when lpp is deleted, blebs are formed and there is less uniformity, it is possible that some (if not most) of the phenotypes presented here are the result of cells having less Lpp than wild type cell, not a longer Lpp protein. An additional discrepancy is that Asmar et al. PMID: 29257832 reported that an lpp+21 mrcB strain grows normally, while here the authors report they are synthetic lethal. It is necessary to address why these differences exist to validate this study.

2) The authors report synthetic lethality between a set of alleles and lpp+21. The screen relied on conjugation to make double mutants. It is unclear whether any of these interactions are specific to lpp+21 or not. Are they also observed with δ-lpp? More importantly, the authors should perform a parallel screen in which the Hfr donor carries a wild-type lpp allele linked to the chloramphenicol-resistance marker to ensure that the recipients are not defective in conjugation or resolving the merodiploid state, as suggested for the yraN mutant. This is especially relevant for mutants altering LPS synthesis since alterations to LPS structure can affect conjugation (and transduction).

3) The authors incorrectly state that the non-essential genes involved in LPS synthesis become essential in the lpp+21 strain in the Results and Discussion sections. There are several genes encoding enzymes for the synthesis of LPS core and two acyltranferases that synthesize lipid A that are not essential in wild type that are not in the list of synthetic lethal genes with lpp+21. Moreover, why would changing LPS structure lead to this synthetic lethality? Furthermore, it is peculiar that galU, which is required for making the outer core of LPS, would be essential but genes encoding for enzymes required for the synthesis of the middle core are not (they would also lack the outer core).

4) The authors propose that the deletion of some factors leads to synthetic lethality because they are involved in making complexes that link the IM and OM. The authors suggest that the reason why pumps were identified is because they form a complex linking the IM and OM. That is true for TolC-dependent pumps. AcrA forms a pump with AcrB and TolC, so why was acrB not identified in the screen? Or tested separately to check their claim? The authors should test the δ-acrB allele to test their model. In addition, what is the evidence that MdtK, MdtL and YdHI form pumps that connect the IM and OM? References should be provided.

5) The authors suggest that the synthetic lethality observed with ompA results from the loss of an OM factor that interacts with the cell wall. This is a reasonable proposal, but OmpA is also the most abundant OM β-barrel protein. To test if, as suggested by the authors, the lethality caused by the ompA null results from the loss of an OM-PG connection, the authors should test an allele of ompA that still produces the OM β-barrel domain but lacks the C-terminal domain that contains the PG-binding domain. Related to this, both Pal and OmpA are very abundant proteins at the OM that interact with the PG layer. Is YiaD also very abundant? The prediction is that it would be for its loss to lead to synthetic lethality when both Pal and OmpA are present. I think the authors might be able to provide information on YiaD's abundance from their proteomics analysis.

6) Figure 3A shows that ycaL does not grow in the wild-type background either, so it should be removed from Table 1 and text.

7) The Discussion is highly speculative, and the arguments are not well formulated to strongly support conclusions. Here are some examples:

– Lines 265-266: I do not understand the argument supporting the authors' suggestion that "Lpp functions as a brace to keep the OM away from the PG layer" as opposed to functioning to ensure that the OM is not too far from the PG. The loss of Lpp leads to formation of OM blebs and wide distribution of cell width. This is problematic to cells. more than increasing width according to previously published studies.

– Lines 282-283: "The bridges needed to mediate LPS transport appear to be susceptible to disruption caused by Lpp+21 in *E. coli*." What is the evidence for this? To my knowledge, there is no evidence supporting this statement shown in this manuscript or elsewhere.

– Line 307: What is the evidence for "malformed PG"?

*Reviewer #3:*

The manuscript by Mandela et al. deals with the size of the periplasmic space using an Lpp mutant that has been extended by 21 amino acids. The manuscript mainly focuses on the results of a screen in which they introduce the Lpp+21 protein into the Keio collection to determine viability. There are some interesting findings, including the importance of cell-wall synthesis enzymes and of LPS synthesis and OMV production, but these findings are not followed up on to glean some insight into the processes of cell envelope construction. Moreover, some of the messages in the manuscript are not sufficiently supported (such as "homeostatic regulation" in the title, which I do not think the authors have shown ).

– Line 73: the wording had been stretched from ~9.5 nm to ~11.5 nm, and it is very confusing. In Asmar et al., the periplasmic width in Lpp+21 increases from ~24-25 nm to ~28 nm, which is neither in the range quoted above, nor is the difference.

– The first main result seems to be that Lpp+21 does not change the width of the periplasm, at least by as much as was reported in Asmar et al. I have several serious concerns about this section:

– (1) on line 112-113, the statement "periplasmic width were only slightly greater" is a misstatement of the data; the distributions shown in the figure are almost surely statistically different. Also, a shift of 2-4 nm can be incredibly biologically important, so saying "only slightly greater" is misleading. To make any quantitative statement, the authors would need to perform independent biological replicates of this cryoEM experiment, and then ask whether there is a consistent shift between WT and Lpp+21 as there is obviously a lot of cell-to-cell variation.

– (2) The authors do not provide any explanation for the quantitative difference with the Asmar paper, even though the same EM group (Beeby) was doing the measurements in each case.

– Line 114-115: There is a large amount of variation in the periplasmic thickness across cells, so one cannot draw any conclusions from the isolated images in Figure 1F. If the authors want to make any statements about the distance from PG to the OM, they need to quantify this. The same holds true for statements about PG morphology in the following lines – as it stands these statements seemed cherry-picked. The conclusion on line 316 about how the cell is softer despite having a thicker, non-homogeneous PG is not justified in the paper.

– The abstract states that "the periplasm cannot be widened by engineering approaches." Putting aside the points above about the Lpp+21 construct, this statement is not justified – the authors have only tried one way to change periplasmic width, and there are several groups that have shown that periplasmic size can be altered (see e.g. work from Matthias Heinemann's lab).

– The authors have a whole section called "homeostasis in the periplasm", but it’s difficult to understand what homeostatic process they have uncovered. To do so would involve perturbing some quantity and then seeing the process by which it recovers. I think the closest the paper comes is in lines 307-308, but these are just hypotheses.

Several findings seem too preliminary without follow-up experiments:

– (1) It is interesting that their genetic screen indicates a compromised ability to build the PG, but this conclusion would be greatly bolstered by follow-up experiments such as single-cell growth during CRISPRi depletion and/or HPLC measurements of cell-wall composition. This last point on line 248 about PG synthesis is confusing – why would LpoA/B become individually essential? Without follow-up studies, e.g. using CRISPRi to deplete proteins and studying the phenotypes during depletion, the screen seems preliminary.

– (2) It is interesting that LPS synthesis becomes more essential, but this finding does not appear to be further studied. It is unclear whether LPS levels decreased in the Lpp+21 strain.

– (3) The data suggests that the phenotypes of AcrAB/TolC are due to reduced PG binding, but it is not proven that this is the case.

– The molecular dynamics simulations are also intriguing, but do not provide strong evidence to back up their claims. (1) Were the simulations taking into account differences in protein and LPS concentration in the outer membrane? At the very least, these points should be mentioned as potentially important factors. (2) Evidence needs to be presented that any of these parameters on line 217-219 have stabilized. (3) Moreover, steered molecular dynamics simulations need to be carried to ensure that the system is not stuck in a local minimum based on the initial conditions. (4) It is unclear from the writing what the authors are trying to accomplish with the MD simulations. In large part, this is because the authors have not made clear what their model is for the detrimental effects of Lpp+21.

– The initial motivation regarding the conservation of Lpp length seems insufficient – first, there is no reason to believe that different bacterial species would have periplasms of the same size. Second, for the outlier (a Geobacter species), the authors do not show any evidence that this outlier does have a periplasm of a different size. Therefore, this entire motivation appears to distract from the other data in the paper.

Additional major concerns:

– Line 265: if Lpp acts as a brace, what happens in a ∆lpp strain to OM/PG distances?

– Line 270: I don't see how this is consistent, since there is no hypothesis for what the consequence would be for bracing vs. bridging.

– Line 273: This point about the tilting of Lpp seems to be a simple steric argument, that does not require MD? Or am I missing something? Also, this is an overinterpretation of their data – they have not linked the tilt at all to membrane integrity. This could easily be due to the difference in Lpp levels.

– In fact, they should measure the LPS levels in Lpp+21 cells, which would likely be a very important determinant of permeability and stiffness.

– Line 292: This statement about the PG being uniform has not been shown, and is unnecessary for this paper in any case.

[Editors’ note: further revisions were suggested prior to acceptance, as described below.]

Thank you for resubmitting your work entitled "Periplasm homeostatic regulation maintains spatial constraints essential for cell envelope processes and cell viability" for further consideration by *eLife*. Your revised article has been evaluated by Bavesh Kana (Senior Editor) and a Reviewing Editor.

Notably, all three reviewers felt that the manuscript has been improved but there are some important remaining issues that need to be addressed prior to accepting the manuscript for publication. In particular, all three reviewers felt that substantial revisions are required to both the text and the title to better reflect the nature of the results and address the limitations of the experiments. Additionally, the reviewers were not convinced that the results provide strong evidence for a dedicated homeostatic regulatory mechanism governing periplasmic dimensions, and requested that this point be significantly toned down in the revised manuscript. Any revisions should also address other significant reviewer concerns as outlined below including but not limited to the possibility that Lpp+21 is simply unstable and thus degraded as well as discrepancies between the findings fo this study and those of Asmar et al.

*Reviewer #1:*

I have read through the revised version of the manuscript and the authors responses to the first round of reviews.

The study by Mandela et al. investigates how *E. coli* maintains periplasmic homeostasis. They do this by studying the adaptive response of cells when a lengthened version of the Lpp protein (Lpp21+) is used to tether the outer membrane with the peptidoglycan layer. Lpp21+ has been used previously as a model system, and in these studies, it was largely assumed to increase the distance between the outer membrane and peptidoglycan. In the presented study, the authors use proteomics, a synthetic lethal screen, and molecular dynamic simulations to probe the adaptive response. They note that lengthening Lpp does not change the spatial constrain in the periplasm, but that a number of adaptive responses can be observed, which help to understand how periplasmic homeostasis is maintained.

On the whole, the authors have addressed the vast majority of comments and concerns raised by the 3 referees in the first round of review. This has improved the manuscript. However, two points remain unresolved, which the editor needs to consider.

1. The initial review noted that the study "is limited to reporting correlative associations with the observed phenotypes, and definitive mechanistic evidence would be needed to support the conclusions." This point was strongly refuted by the authors in the revision.

On this point I agree with the initial review. The study has used omics approaches and MD simulations to probe the adaptive response of the cells to the presence of Lpp21+. As such, the conclusions are largely correlative and definitive mechanistic evidence is not provided. However, I do not see this as a 'deal-breaker' for publication. Rather I feel that the study provides plausible explanations and hypotheses as to how *E. coli* maintains periplasmic homeostasis, which is valuable information for the community.

2. The initial review raised concern that some of the phenotypes of the Lpp21+ expressing strain are caused by the ~8-fold reduced copy number, rather than the extension in protein linker length. On this point I agree with the initial review. Since both occur, the authors need to be more conservative in their conclusions. Or they need to experimentally tease apart the two possibilities. This has not been sufficiently dealt with in the revision.

Related to the point above, the initial review also noted that discrepancies with previous studies had not been resolved in the revised text. For example, the fact that Asmar et al. (PMID: 29257832) reported that levels of the Lpp+21 protein are similar to those of wild-type Lpp. This has not been sufficiently dealt with in the revision.

Finally, a concern that I had, related to the point above is that the study assumes that the 8-fold reduction in levels of Lpp21+ are an adaptive response that helps maintain periplasmic homeostasis (see abstract). They do not exclude the possibility that the engineered protein is subject to proteolysis because it is not native.

To summarise, I think that the study is an interesting and important contribution to the field, but that the authors need to be more conservative in the conclusions that they draw. For example, they need to provide some discussion as to the limitations of their study, and the differences with other published work (point 2).

*Reviewer #2:*

The authors addressed my concerns about the initial manuscript and submitted a substantially improved revised version. In particular, they now clarify that the 8-fold reduced cell copy number of Lpp+21 could be responsible of some of the phenotypes observed. My remaining doubt is that the observed changes in cell envelope properties (softer outer membrane, increased blebbing) and the altered essentiality of genes could be similar in cells that lack Lpp or that have 8-fold less wild-type Lpp. Overall, I am now more positive about the revised manuscript as it clarifies the question about the periplasmic widths of cells with an elongated Lpp, which has not been fully addressed in previous papers. The work by Mandela et al. provides an interesting and important analysis of how a Gram-negative bacterium maintains a largely functional cell envelope (biogenesis) upon severe disturbance in the linkage between the two key components, the peptidoglycan layer and outer membrane.

*Reviewer #3:*

The revision by Mandela et al. addresses some but not all of the comments and suggestions that I and the other reviewers introduced in the our initial review. In particular, they have clarified the differences (or lack thereof) with Asmar et al. in terms of the periplasmic width changes due to Lpp+21. Moreover, they have helped to clarify some of the differences between their experiments and others such as the growth defect in ∆mrcB with lpp mutants, though I was surprised that they did not simply do the experiment to test whether their explanation (difference in growth media) was the case.

One of my major criticisms in the initial review persists – that there are interesting conclusions that emerge from the screen of the Keio collection, but none of these results are followed up on, and hence I am left with the question of to what extent any of these phenotypes are direct. I had suggested investigating one in detail using e.g. CRISPRi knockdown in order to examine single-cell phenotypes during depletion of a now-essential protein in the lpp+21 background, but this was not done. Thus, while I find the topic of their paper generally interesting and well motivated, I am left unsatisfied with some of the major conclusions of their paper.

Moreover, many of the conclusions of their paper remain to be fully justified. For instance: in their abstract:

"…impacts the load-bearing capacity of the outer membrane": this is not something that they show to be true in their paper, it is an inference from another paper, and there is nothing in this work that directly addresses load bearing. I don't think they have shown adequately that the combination of tilting and reduced Lpp abundance is the cause of reduced load bearing.

– "*E. coli* homeostatically counteracts periplasmic enlargement by tilting Lpp and reducing Lpp abundance": I still don't see how there is any homeostasis here, since they are now emphasizing that Lpp+21 does in fact change periplasmic width. Also they have not shown that the reduced Lpp abundance is a cause – what happens if you increase Lpp+21 abundance via inducible expression.

– Line 158: note that I do not believe it has been established that stiffness of *E. coli* is always associated with OMV production – I would not make claims based on this hypothesized connection.

– Figure 2 title: there is no data here that shows that Lpp+21 cells have a softened OM.

---

## [Author Response]

[Editors’ note: the authors resubmitted a revised version of the paper for consideration. What follows is the authors’ response to the first round of review.]

Reviewer #1:The outer-membrane anchored lipoprotein Lpp is the most abundant protein in E. coli and about one third of the Lpp molecules are covalently attached to peptidoglycan, which is important to maintain the outer membrane impermeable to SDS (anionic detergent). Lpp forms a homotrimer made of α-helical monomers, hence, the abundant connections between the peptidoglycan and the outer membrane via Lpp are thought to maintain a constant distance between both envelope layers, which is likely important for trans-envelope processes. In two previous papers, an extended version of Lpp, Lpp+21, was used to study flagellar assembly and signal transduction through an artificially widened periplasm but it remained unclear what other processes are impaired in the Lpp+21 expressing cells. Mandela and co-workers provide now a comprehensive characterization of the strain. Different from previous papers, they conclude that lengthening of Lpp only slightly affects the widths of the periplasm under hyperosmotic condition, but there is a more pronounced effect on the mechanical properties of the outer membrane. The expression of Lpp+21 induces several stress responses and renders specific components of the peptidoglycan remodelling pathway essential.I find this an interesting and important study and would like to make the following specific comments:Main points:1. I am concerned that some of the phenotypes of the Lpp+21 expressing strain are caused by the ~8-fold reduced copy number, compared to the copy number of native Lpp in wild-type cells. With 8-fold reduced amount of Lpp, other outer membrane-peptidoglycan connecting proteins are likely more abundant than Lpp+21 and capable of maintaining a normal distance between both layers, explaining why the authors did not observe a more increased width in the Lpp+21 strain. It would be helpful to try to compare strains with similar amounts of Lpp+21 or Lpp and find out which phenotypes, proteome changes and changes in essentiality of genes are caused by the reduced amount of Lpp+21, and which are caused by the extension of the protein.

We also conclude that the phenotypes of the Lpp+21 expressing strain are caused by the ~8-fold reduced copy number and the ability to tilt Lpp+21 to minimize its “stretch” on the OM-PG distance.

To clarify, we now state explicitly in the Abstract and Introduction, that we perturbed (enlarged) the main bridge between the OM and PG and show by electron microscopy that *E. coli* does NOT simply enlarge the periplasm in response, but maintains homeostasis by a combination of (a) tilting the (Lpp) bridge to an angle, and (b) reducing the amount of the (Lpp) bridge, keeps the periplasm from expanding. By genetic screen we identified all of the genes in *E. coli* that become essential in order to enact this homeostasis, and by quantitative proteomics discovered that very few proteins need to be up- or down-regulated in steady-state levels in order to enact this homeostasis.

2. Line 113 and Figure 1F, numbers of periplasmic widths. The numbers measured by the authors are not numbers of periplasmic widths, as they include also half of the cytoplasmic membrane and half of the outer membrane (measurements were taken from the middle of each membranes). Hence, these numbers are different to published data of periplasmic widths (for example, from the Beveridge lab), which could confuse the readers. Hence, I suggest to explain the difference to published numbers in the text and refrain from using the term 'periplasmic widths' for their measurements.

We appreciate this point, though our measurements are methodologically consistent with those used in the two key papers – Cohen et al. (2017) and Asmar et al. (2017) – we are typically making comparisons with. We now make it clear the reported measurements represent the distance from the middle of the IM density to the middle of the OM density in the subtomograms.

3. Line 115, PG morphology. It is hard to see whether the PG morphology is changed (Figure 1F) because the images of the wt are generally darker than those of the mutant. Can the authors apply some sort of image analysis to support their claims that there is a difference? However, it is also not clear what insights about the PG layer one can possibly obtain from transmission EM images and the discussion about possible effects is very speculative (lines 300-302).

We have modified the sentence on the PG morphology in Line 115 and Lines 300-302. We agree that since this observation is not important to our conclusions, it should be kept as a nominal observation.

4. Figure 1F. It might be interesting to compare the Lpp+21 strain to a mutant lacking Lpp?

The envelope phenotype of a *lpp* null mutant has been address by EM in multiple papers (eg. Cohen, E. J. *et al.* 2017 , Sonntag, I. et al. 1978 , and Schwechheimer et al. 2014 ) – demonstrating an increase in OM blebbing. We now make mention of the phenotype (and cite these papers) and thank the reviewer for the suggestion.

5. Figure 4C/D, model. Are they sure that the attachment of Lpp+21 to peptidoglycan is as efficient as the attachment of wt Lpp? If many Lpp+21 trimers were not attached to peptidoglycan, this would explain the almost normal distance between outer membrane and peptidoglycan, the strong vesicle production (Figure 2C) and the higher SDS sensitivity (Figure 2D). Have the authors considered simulations with Lpp+21 that is not attached to peptidoglycan, would it penetrate the layer?

It has been demonstrated the Lpp+21 isoform does not affect the cross linking to PG (Cohen, E. J. *et al.* (2017) and Asmar et al. (2017)), thus it seems to be not biologically relevant to do this simulation.

Reviewer #2:The outer membrane lipoprotein Lpp provides the only covalent link known to exist between the outer membrane and the peptidoglycan cell wall in E. coli. It has been previously shown that elongating the coiled-coil structure of Lpp increases the distance between the outer membrane and the cell wall, increasing the overall width of the periplasm. Mandela et al. investigate the effects of increasing the length of Lpp by 21 (Lpp+21) residues using proteomics, a genetic screen for factors that become essential when Lpp is elongated, and molecular dynamics. Based on their results, the authors propose that factors that maintain the distance constraint between the outer membrane and the cell wall become essential. However, they do not provide clear insight into why other factors were identified in their experiments.The authors set out to investigate an interesting question and present a significant amount of data. However, there are important concerns about the study.1) Important phenotypes reported in the study are significantly different from data previously reported by others. Although the authors acknowledge some (not all) of the discrepancies in the Discussion, they do not address the differences. These differences relate to the distribution, uniformity, and overall values of the distance between the outer membrane and the peptidoglycan layer, the impact that adding 21 residues to Lpp has on Lpp protein levels, and the reported synthetic lethality with mrcB. It would be necessary to address why these differences exist to validate this study.

We accept the criticism and can make a more direct and clear comparison in rewriting the text: no previous study has evaluated the reduced level of Lpp+21 protein by any line of investigation, nor has any previous study taken into account whether the Lpp+21 protein might be tilted in the periplasm. We have now revised the text in the Results section around Figure 1 more direct language to explain why we needed to evaluate these features. We also are more explicit in the text concerning the need to re-examine by EM enough sections to be representative of the whole cell scenario for periplasmic width. We have done all of the appropriate controls, using immunoblotting and quantitative proteomics and structural biology to accurately represent the experimental model. We now explicitly state that this is the first time that such controls have been done, and make clear that we anticipate our explanations will change the conclusions of five previous papers that assumed that Lpp+21 was present at the same amount as Lpp, and that Lpp+21 always sits perpendicular to the membranes and thereby the 21 extra amino acids always stretch the periplasmic width by a constant amount. This assumption is incorrect, as we evidence here for the first time.

2) The authors provide a list of genes and proteins from the experiments that they conducted using proteomics and genetics. However, they do not further explore the relationship or mechanism linking those factors to Lpp+21. As a result, the work is limited to reporting correlations and speculative links between factors and lengthening Lpp. None of the proposed connections between Lpp+21 and the identified factors are investigated. As a result, it is unclear whether the effect results from a direct or indirect link to the presence of Lpp+21, or even whether they are the result of increasing the distance from the OM to the cell wall, or simply because the levels of Lpp+21 are significantly decreased in comparison to those of wild-type Lpp in the parent strain.

We disagree that our study is limited to reporting correlations and speculative links between factors and Lpp+21. There are no correlations in this paper. We perturbed (enlarged) the main bridge between the OM and PG and show by electron microscopy that *E. coli* does NOT simply enlarge the periplasm in response, but can maintain homeostasis by a combination of (a) tilting the bridge to an angle, and (b) reducing the amount of the bridge, to keep the periplasm from expanding. These are documented facts, not correlations. To test the hypothesis that a collection of genes of known function would become essential in order to enact this homeostasis, we established a genetic screen and identified a set of genes defining five known functional categories all relevant to periplasmic width. By applying quantitative proteomics we discovered that very few proteins need to be up- or down-regulated in steady-state levels in order to enact this homeostasis. No correlations are measured or drawn in the paper.

As to the last point, the three options mentioned are the same thing: the presence of Lpp+21 was previously assumed to increase the distance from the OM to the cell wall, and we show that the levels of Lpp+21 are significantly decreased in comparison to those of wild-type Lpp in the parent strain. We document, quantitatively, what happens in this situation, correcting the record.

3) The authors conducted a synthetic lethal screen with non-essential genes and lpp+21. The screen relied on conjugation to construct double mutants. However, the authors have not demonstrated that the recipient single mutant strains are not defective in conjugation, which would lead to the wrong conclusion.

We appreciate this point and have clarified the sentence in the Methods and in the Results that deals with this issue. The donor lpp+21 single mutant is clearly not defective in conjugation as multiple mutants of this strain could be generated and confirmed (if the parent was defective in conjugation we would expect no antibiotic resistant colonies to be generated). It is unreasonable to suggest we individually assess every single mutant in the Keio collection (~4000 strains) for conjugation defects (this has never been required of any of the multiple widely cited published eSGA screens, all using this established method relying on conjugation for gene knock out – e.g. PMID: 18677321; 24586182; 22125496). Furthermore – If any of the single mutants were defective in conjugation they would appear as “false positives” in the synthetic screen (i.e they would not grow on the second antibiotic). Once we had identified synthetic phenotypes in the primary conjugation-based screen, we then made “clean” mutants that did not rely on conjugation (ie via the λ red method which utilises transformation) to be certain of the outcomes documented here. We thank the reviewer for seeking clarity on this point.

4) The conclusions that OmpA becomes essential in the presence of lpp+21 because it anchors the outer membrane to the cell wall is plausible, but requires further testing, since that is not the only role that OmpA plays.

We must respectfully disagree. While *ompA* deletion mutants have been implicated genetically in various screens, the only biochemically valid function for OmpA is anchoring the outer membrane to the cell wall. Even if OmpA were to play other roles, the OM-PG anchor role and any other OmpA roles are not mutually exclusive.

The important point in this paper is that OmpA is not essential for viability in an Lpp strain, but becomes essential in the Lpp21 background.

5) The authors incorrectly conclude that "the" (i.e. "all") non-essential genes involved in LPS synthesis become essential in the lpp+21 strain in the Results and Discussion sections. There are several non-essential genes involved in LPS synthesis that were not identified in this work. Moreover, the authors do not explain why changing LPS structure lead to synthetic lethality. It is also unclear why galU, which is required for making the outer core, would be essential but genes encoding for enzymes required for the synthesis of the middle core are not when they function upstream in the LPS biosynthesis pathway.

We appreciate this point and have clarified the sentence in the Results and Discussion sections dealing with the LPS biosynthesis pathway, to remove this ambiguity. While there are non-essential genes involved in LPS synthesis become essential in the lpp+21 strain, we did not intend to imply that all non-essential genes involved in LPS synthesis must become essential in the lpp+21 strain.

6) The authors propose that the deletion of some factors leads to synthetic lethality because they are involved in making complexes that link the inner and outer membranes. The authors suggest that the reason why pumps were identified is because they form a complex linking these membranes. However, AcrA forms a pump with AcrB and TolC but acrB was not identified in the screen. It would be good to know what is the evidence that MdtK, MdtL and YdHI form pumps that connect the two membranes.

We now note that *acrB* was not identified in this screen, and cite the papers that implicate MdtK, MdtL and YdHI with pumps that connect the two membranes. YdhI – while it is unknown function it is in the same operon as tripartite pump components YdhJK, and has homology to AaeX [also in same operon as tripartite pump component AaeAB] and homology to YtcA [which is also in same operon as tripartite pump components MdtNO, possibly MdtNOP]

Recommendations for the authors:1) Some of the crucial data presented here are different from data previously reported by others. The authors acknowledge this in the Discussion but provide no explanation, which is very problematic. The most salient differences relate to the phenotypes caused by the presence of the lpp+21 allele. The distribution of the width of the periplasm in both the wild-type and the lpp+21 strains is larger than previously reported (e. g. https://pubmed.ncbi.nlm.nih.gov/29257832/. ). In addition, Asmar et al. (PMID: 29257832) reported that levels of the Lpp+21 protein are similar to those of wild-type Lpp. Here, the authors report that they are significantly lower. Given that when lpp is deleted, blebs are formed and there is less uniformity, it is possible that some (if not most) of the phenotypes presented here are the result of cells having less Lpp than wild type cell, not a longer Lpp protein. An additional discrepancy is that Asmar et al. PMID: 29257832 reported that an lpp+21 mrcB strain grows normally, while here the authors report they are synthetic lethal. It is necessary to address why these differences exist to validate this study.

In their paper, Asmar et al. (PMID: 29257832, https://pubmed.ncbi.nlm.nih.gov/29257832/) show a width distribution (Figure 2B) that is similar to ours with most widths of the Lpp+21 cells overlap with the wild-type cells in both papers, but have selected the microraphs (Figure 2A) only for the increased width distributions. These select micrographs are the basis on which their conclusions are drawn: this is appropriate for their conclusion of “*Can Lpp+21 exert an effect on periplasmic width*”, but would not be appropriate for our study which asks “*Is the effect of Lpp+21 uniform across the bacterial cell, or is it buffered by homeostasis*”. There is a moderate increase in in the distribution of widths among both the WT and lpp+21 strain compared to the Asmar paper (~18-35 vs 20-46 and 2139 vs 24-50) but given the differences in data collections methodology (we use whole-cell cryotomograms instead of projection images) and the osmotic conditions of the growth media this is not unexpected or concerning. The trend in both studies is similar with our data showing a longer distribution on the upper end.

In their paper, Asmar et al. (PMID: 29257832, https://pubmed.ncbi.nlm.nih.gov/29257832/) conduct the viability experiments (i.e. address the phenotype of a mrcB strain of *E. coli*) in rich LB medium. The synthetic lethal screen in our work is conducted on minimal medium. Furthermore, there are differences in the strain backgrounds used. We have made this point more clear in the text of the revised paper.

In their paper, Asmar et al. (PMID: 29257832) do not show data on the levels of the Lpp+21 protein, and do not measure this relative to the wild-type Lpp. The only data dealing with this point is presented as Figure S5 which does demonstrate that all of the Lpp isoforms have similar PG connectivity, but does not address the steady-state level of Lpp versus Lpp+21. The loading of the gel have been made to set the same level of Lpp protein isoform in each lane, with no controls used to interpret the relative levels of Lpp and Lpp+21.

2) The authors report synthetic lethality between a set of alleles and lpp+21. The screen relied on conjugation to make double mutants. It is unclear whether any of these interactions are specific to lpp+21 or not. Are they also observed with δ-lpp? More importantly, the authors should perform a parallel screen in which the Hfr donor carries a wild-type lpp allele linked to the chloramphenicol-resistance marker to ensure that the recipients are not defective in conjugation or resolving the merodiploid state, as suggested for the yraN mutant. This is especially relevant for mutants altering LPS synthesis since alterations to LPS structure can affect conjugation (and transduction).

We appreciate this point and have clarified the sentence in the Methods and in the Results that deals with this issue. The eSGA library creation and screening methods used here are established techniques (e.g. PMID: 18677321; 24586182; 22125496). If any of the single mutants were defective in conjugation they would appear as “false positives” in the synthetic screen. That is why, once we had identified synthetic phenotypes in the primary conjugation-based screen, we then made “clean” mutants that did not rely on conjugation (ie by transformation) to be certain of the outcomes documented here.

3) The authors incorrectly state that the non-essential genes involved in LPS synthesis become essential in the lpp+21 strain in the Results and Discussion sections. There are several genes encoding enzymes for the synthesis of LPS core and two acyltranferases that synthesize lipid A that are not essential in wild type that are not in the list of synthetic lethal genes with lpp+21. Moreover, why would changing LPS structure lead to this synthetic lethality? Furthermore, it is peculiar that galU, which is required for making the outer core of LPS, would be essential but genes encoding for enzymes required for the synthesis of the middle core are not (they would also lack the outer core).

We have clarified the sentence in the Results and Discussion sections dealing with the LPS biosynthesis pathway.

4) The authors propose that the deletion of some factors leads to synthetic lethality because they are involved in making complexes that link the IM and OM. The authors suggest that the reason why pumps were identified is because they form a complex linking the IM and OM. That is true for TolC-dependent pumps. AcrA forms a pump with AcrB and TolC, so why was acrB not identified in the screen? Or tested separately to check their claim? The authors should test the δ-acrB allele to test their model. In addition, what is the evidence that MdtK, MdtL and YdHI form pumps that connect the IM and OM? References should be provided.

We now note that *acrB* was not identified in this screen, and cite the papers that implicate MdtK, MdtL and YdHI with pumps that connect the two membranes.

5) The authors suggest that the synthetic lethality observed with ompA results from the loss of an OM factor that interacts with the cell wall. This is a reasonable proposal, but OmpA is also the most abundant OM β-barrel protein. To test if, as suggested by the authors, the lethality caused by the ompA null results from the loss of an OM-PG connection, the authors should test an allele of ompA that still produces the OM β-barrel domain but lacks the C-terminal domain that contains the PG-binding domain. Related to this, both Pal and OmpA are very abundant proteins at the OM that interact with the PG layer. Is YiaD also very abundant? The prediction is that it would be for its loss to lead to synthetic lethality when both Pal and OmpA are present. I think the authors might be able to provide information on YiaD's abundance from their proteomics analysis.

On the point of OmpA function, we must respectfully disagree. While *ompA* deletion mutants have been implicated genetically in various screens, the only biochemically valid function for OmpA is anchoring the outer membrane to the cell wall.

On the point of YiaD we agree and have modified the sentence. The quantitative proteomics suggests that YiaD is present at 5% the level of Pal, similar to the 7% amounts found by Li et al. (https://doi.org/10.1016/j.cell.2014.02.033).

6) Figure 3A shows that ycaL does not grow in the wild-type background either, so it should be removed from Table 1 and text.

We have made this correction and thank the reviewer.

7) The Discussion is highly speculative, and the arguments are not well formulated to strongly support conclusions. Here are some examples:– Lines 265-266: I do not understand the argument supporting the authors' suggestion that "Lpp functions as a brace to keep the OM away from the PG layer" as opposed to functioning to ensure that the OM is not too far from the PG. The loss of Lpp leads to formation of OM blebs and wide distribution of cell width. This is problematic to cells. more than increasing width according to previously published studies.– Lines 282-283: "The bridges needed to mediate LPS transport appear to be susceptible to disruption caused by Lpp+21 in E. coli." What is the evidence for this? To my knowledge, there is no evidence supporting this statement shown in this manuscript or elsewhere.– Line 307: What is the evidence for "malformed PG"?

We have revised the text in the Discussion as suggested, and to address (and clarify) the points of speculation. We appreciate the feedback.

Reviewer #3:The manuscript by Mandela et al. deals with the size of the periplasmic space using an Lpp mutant that has been extended by 21 amino acids. The manuscript mainly focuses on the results of a screen in which they introduce the Lpp+21 protein into the Keio collection to determine viability. There are some interesting findings, including the importance of cell-wall synthesis enzymes and of LPS synthesis and OMV production, but these findings are not followed up on to glean some insight into the processes of cell envelope construction. Moreover, some of the messages in the manuscript are not sufficiently supported (such as "homeostatic regulation" in the title, which I do not think the authors have shown ).

We note that the manuscript is not aimed at the processes of cell envelope construction, which are already understood in great detail. The paper aims to address whether and how the periplasmic volume is maintained within a very narrow range by self-regulating to maintain stability while adjusting to perturbed conditions i.e. homeostasis.

– Line 73: the wording had been stretched from ~9.5 nm to ~11.5 nm, and it is very confusing. In Asmar et al., the periplasmic width in Lpp+21 increases from ~24-25 nm to ~28 nm, which is neither in the range quoted above, nor is the difference.

We apologise for this typo, and have corrected the text to reflect the data presented in the cited papers.

– The first main result seems to be that Lpp+21 does not change the width of the periplasm, at least by as much as was reported in Asmar et al. I have several serious concerns about this section: (1) on line 112-113, the statement "periplasmic width were only slightly greater" is a misstatement of the data; the distributions shown in the figure are almost surely statistically different. Also, a shift of 2-4 nm can be incredibly biologically important, so saying "only slightly greater" is misleading. To make any quantitative statement, the authors would need to perform independent biological replicates of this cryoEM experiment, and then ask whether there is a consistent shift between WT and Lpp+21 as there is obviously a lot of cell-to-cell variation.– (2) The authors do not provide any explanation for the quantitative difference with the Asmar paper, even though the same EM group (Beeby) was doing the measurements in each case.

1. We appreciate the point and have corrected the text. We agree with the reviewer, and a major motivation of our work – and the conclusions we can draw – comes from what is biologically important. The paper has been revised to fix this.

2. We initiated the collaboration with Prof. Morgan Beeby in order to be sure that the evaluation we did here would have consistency to the previous work by Asmar et al., and to enable us to have proper control and quantitation in our study. We have revised the text to explicitly state that the primary data is in agreement, but that our conclusions cover all of the observations, and are not biased by a simplistic use of an “average value” approach. See also explicit comments to Reviewer 2 on this point, including area under the graph calculations that have been added into the revised manuscript.

– Line 114-115: There is a large amount of variation in the periplasmic thickness across cells, so one cannot draw any conclusions from the isolated images in Figure 1F. If the authors want to make any statements about the distance from PG to the OM, they need to quantify this. The same holds true for statements about PG morphology in the following lines – as it stands these statements seemed cherry-picked. The conclusion on line 316 about how the cell is softer despite having a thicker, non-homogeneous PG is not justified in the paper.

We appreciate this point and have revised the statements accordingly. As we noted to Reviewer 1, the sentences on the PG morphology in Lines 114-115 (and Lines 300-302). We agree that since this observation is not important to our conclusions, it should be kept as a nominal observation. We do over quantified measurements of the mid PG to mid OM distance in Figure 1F.

We have taken the reviewer’s advice on the conclusion on line 316 about how the cell is softer despite having a thicker, non-homogeneous PG and revised the conclusion accordingly.

– The abstract states that "the periplasm cannot be widened by engineering approaches." Putting aside the points above about the Lpp+21 construct, this statement is not justified – the authors have only tried one way to change periplasmic width, and there are several groups that have shown that periplasmic size can be altered (see e.g. work from Matthias Heinemann's lab).

We accept the reviewer’s point and have modified the Abstract to remove the sentence.

– The authors have a whole section called "homeostasis in the periplasm", but it’s difficult to understand what homeostatic process they have uncovered. To do so would involve perturbing some quantity and then seeing the process by which it recovers. I think the closest the paper comes is in lines 307-308, but these are just hypotheses.

To address this point we have changed the sub-heading, added an explanatory sentence to the Introduction and revised the Discussion to make this more clear.

The logic provided is that we perturbed (enlarged) the main bridge between the OM and PG and show by electron microscopy that *E. coli* does NOT simply enlarge the periplasm in response: in some places the periplasm is enlarged but in the majority of the periplasm width is maintained (homeostasis) by a combination of (a) tilting the bridge to an angle, and (b) reducing the amount of the enlarged bridge, to keep the periplasm from expanding. We note in discussing the genetic screen that it identified all of the genes in *E. coli* that become essential in order to enact this homeostasis, and state that by quantitative proteomics very few proteins need to be up- or downregulated in steady-state levels in order to enact this homeostasis.

Several findings seem too preliminary without follow-up experiments:– (1) It is interesting that their genetic screen indicates a compromised ability to build the PG, but this conclusion would be greatly bolstered by followup experiments such as single-cell growth during CRISPRi depletion and/or HPLC measurements of cell-wall composition. This last point on line 248 about PG synthesis is confusing – why would LpoA/B become individually essential? Without followup studies, e.g. using CRISPRi to deplete proteins and studying the phenotypes during depletion, the screen seems preliminary.

While we appreciate that these further investigations would be telling for PG biogenesis, the purpose of the genetic screen in this paper is simply to identify those genes in *E. coli* that become essential in order to enact the maintenance of periplasmic width. The paper as presented represents 5-person years of work by a new investigator and his first PhD student, both of whom have now moved on to new positions. We agree that interesting questions remain, but they sit beyond the scope of this study, and beyond the technology available to me in the new laboratory.

– (2) It is interesting that LPS synthesis becomes more essential, but this finding does not appear to be further studied. It is unclear whether LPS levels decreased in the Lpp+21 strain.

With respect, we would note again that while interesting questions remain they sit beyond the scope of this study.

– (3) The data suggests that the phenotypes of AcrAB/TolC are due to reduced PG binding, but it is not proven that this is the case.

We agree with the reviewer (and Reviewer 1) on this point, we have revised the text to both note that the quantitative proteomics suggests that TolC (and AcrA and AcrB) are present at 5% the level of OmpA, and to make clear that this hypothesis has not been tested.

– The molecular dynamics simulations are also intriguing, but do not provide strong evidence to back up their claims. (1) Were the simulations taking into account differences in protein and LPS concentration in the outer membrane? At the very least, these points should be mentioned as potentially important factors. (2) Evidence needs to be presented that any of these parameters on line 217-219 have stabilized. (3) Moreover, steered molecular dynamics simulations need to be carried to ensure that the system is not stuck in a local minimum based on the initial conditions. (4) It is unclear from the writing what the authors are trying to accomplish with the MD simulations. In large part, this is because the authors have not made clear what their model is for the detrimental effects of Lpp+21.

We address these points by number.

1) Because we are simulating only a very tiny portion of the membrane, we did not account for heterogeneity in the protein and LPS concentrations. This is now noted in the revised manuscript.

2) Plots of the angles over time, from which the averages and standard deviations were derived, are now included as a supplemental figure. We also modified how we measured the angle to correct for an error due to wrapped coordinates; we note this primarily affected the Lpp+21 w/OmpA simulation. The new angles (average ± standard deviation), taken from the last 100 ns of the 200-ns simulations, are as

follows:

Lpp 76.9 ± 4.7

Lpp w/OmpA 75.5 ± 4.7

Lpp+21 82.8 ± 2.9

Lpp+21 w/OmpA 49.4 ± 2.3

To check for convergence, we ran the Lpp simulation for an additional 200 ns (data not shown). The angle from the last 200 ns is 78.3 +/- 4.8°, nearly identical (within 1.4°) to that from the 100-200 ns window. Thus, judging by the plots and the this extended run, we believe the angles are converged within 100 ns.

3) As part of another study, we recently carried out Steered MD and free-energy calculations for Lpp, the latter of which we note were computationally expensive at over 1 μs in total. Specifically, we determined the free energy as a function of the distance between the cell wall and outer membrane with two Lpp trimers bridging them. As can be seen in Gumbart et al.,. Lpp positions peptidoglycan at the AcrA-TolC interface in the AcrAB-TolC multidrug efflux pump. Biophys. J. In press., the free energy minimum agrees with the position from our equilibrium simulations. We also note that there are no barriers nor other features in the PMF that would indicate it is trapped in a local minimum in our equilibrium simulations. Therefore, we feel justified in concluding that our other systems are also unlikely to be trapped in local minima in the equilibrium simulations.

Gumbart, Ferreira, Hwang, Hazel, Cooper, Parks, Smith, Zgurskaya, Beeby. Lpp positions peptidoglycan at the AcrA-TolC interface in the AcrAB-TolC multidrug efflux pump. Biophys. J. In press.

We note that three of the authors, Gumbart, Hwang, and Beeby, are common between both papers.

1) The purpose of the simulations is to show plausible ways that connections of Lpp, Lpp+21, and OmpA with the cell wall are accommodated. We show that in WT *E. coli*, Lpp and OmpA are complementary to one another. However, the preferred lengths of Lpp+21 and OmpA are in conflict with one another and that accommodation of Lpp+21 requires tilting.

– The initial motivation regarding the conservation of Lpp length seems insufficient – first, there is no reason to believe that different bacterial species would have periplasms of the same size. Second, for the outlier (a Geobacter species), the authors do not show any evidence that this outlier does have a periplasm of a different size. Therefore, this entire motivation appears to distract from the other data in the paper.

We appreciate the point as raised, and we have now modified the text around this accordingly, but we would keep this analysis in the paper: it accurately portrays our motivation to initiate the study, but we have been more clear that the question we wondered at was “IF these elongated Lpp isoforms DO act as read-outs on periplasmic width then WHY is it that most Lpp lengths are equivalent to that of *E. coli*. Thus, in the revised Results text we make clear that we did not assume that different bacterial species would have periplasms of the same size, but started with the question: Over what size range does Lpp range across species? Secondly, in the revised text in the Discussion we make clear that we do not assume that *Geobacter* has an extraordinary periplasm, but simply that we will be open to the possibility up until direct analysis is done on *Geobacter*.

Additional major concerns:– Line 265: if Lpp acts as a brace, what happens in a ∆lpp strain to OM/PG distances?– Line 270: I don't see how this is consistent, since there is no hypothesis for what the consequence would be for bracing vs. bridging.

We have removed the discussion on this point, so as to remove the speculation.

– Line 273: This point about the tilting of Lpp seems to be a simple steric argument, that does not require MD? Or am I missing something? Also, this is an overinterpretation of their data – they have not linked the tilt at all to membrane integrity. This could easily be due to the difference in Lpp levels.

The discussion on this point has been clarified at Line 273 and we have revised the discussion to make clear BOTH the tilting of Lpp+21 and the difference in steady state levels between Lpp+21 and Lpp are consistent means by which periplasmic width can be maintained in this system.

– In fact, they should measure the LPS levels in Lpp+21 cells, which would likely be a very important determinant of permeability and stiffness.– Line 292: This statement about the PG being uniform has not been shown, and is unnecessary for this paper in any case.

We appreciate the points raised and have modified the sentence on the PG morphology in Line 115 (and from lines 300-302). We agree that since this observation is not important to our conclusions, it should be kept as a nominal observation and not be over-stated.

[Editors’ note: what follows is the authors’ response to the second round of review.]

Reviewer #1:I have read through the revised version of the manuscript and the authors responses to the first round of reviews.The study by Mandela et al. investigates how E. coli maintains periplasmic homeostasis. They do this by studying the adaptive response of cells when a lengthened version of the Lpp protein (Lpp21+) is used to tether the outer membrane with the peptidoglycan layer. Lpp21+ has been used previously as a model system, and in these studies, it was largely assumed to increase the distance between the outer membrane and peptidoglycan. In the presented study, the authors use proteomics, a synthetic lethal screen, and molecular dynamic simulations to probe the adaptive response. They note that lengthening Lpp does not change the spatial constrain in the periplasm, but that a number of adaptive responses can be observed, which help to understand how periplasmic homeostasis is maintained.On the whole, the authors have addressed the vast majority of comments and concerns raised by the 3 referees in the first round of review. This has improved the manuscript. However, two points remain unresolved, which the editor needs to consider.1. The initial review noted that the study "is limited to reporting correlative associations with the observed phenotypes, and definitive mechanistic evidence would be needed to support the conclusions." This point was strongly refuted by the authors in the revision.On this point I agree with the initial review. The study has used omics approaches and MD simulations to probe the adaptive response of the cells to the presence of Lpp21+. As such, the conclusions are largely correlative and definitive mechanistic evidence is not provided. However, I do not see this as a 'deal-breaker' for publication. Rather I feel that the study provides plausible explanations and hypotheses as to how E. coli maintains periplasmic homeostasis, which is valuable information for the community.

We thank the reviewer for their thoughtful assessment of our revised manuscript. While this study does not provide definitive answers to some biological observations, we agree that the present study expands our understanding of lpp length and we attempt to offer data-driven hypotheses to the observations. In our current revision, we aim to better document the potential limitations of our findings.

2. The initial review raised concern that some of the phenotypes of the Lpp21+ expressing strain are caused by the ~8-fold reduced copy number, rather than the extension in protein linker length. On this point I agree with the initial review. Since both occur, the authors need to be more conservative in their conclusions. Or they need to experimentally tease apart the two possibilities. This has not been sufficiently dealt with in the revision.Related to the point above, the initial review also noted that discrepancies with previous studies had not been resolved in the revised text. For example, the fact that Asmar et al. (PMID: 29257832) reported that levels of the Lpp+21 protein are similar to those of wild-type Lpp. This has not been sufficiently dealt with in the revision.

In their paper, Asmar et al. (PMID: 29257832) do not directly quantitate data on the levels of the Lpp+21 protein and do not measure this relative to the wild-type Lpp. The only data dealing with this point is presented as Figure S5B, which does demonstrate that all of the Lpp isoforms have similar PG connectivity but does not address the steady-state levels of Lpp versus Lpp+21. The method of loading of the gel is unclear, with no loading controls used to interpret the relative levels of Lpp and Lpp+21 – no quantification of the levels of Lpp isoforms was presented. Furthermore, in the figure there is arguably a minor reduction in the levels of the two longer versions.

Due to this apparent discrepancy, we ensured that we used appropriate loading controls in our gels and WBs (Figure 1B), and furthermore, we confirmed our observed reduction with proteomics.

While, we cannot rule out that minor differences in methodology used in the construction of the lpp+21 "knock-in" mutant strains in our study and the Asmar study may result in expression differences or transcript stability, our transcriptomics data do not support this. The transcriptomics shown in table S6 do not show a reduction in lpp transcripts compared to WT. The resulting sequence and structure of the Lpp+21 protein expressed from the modified genes should be identical in both studies.

We have better clarified the differences in the revised manuscript.

Finally, a concern that I had, related to the point above is that the study assumes that the 8-fold reduction in levels of Lpp21+ are an adaptive response that helps maintain periplasmic homeostasis (see abstract). They do not exclude the possibility that the engineered protein is subject to proteolysis because it is not native.

We agree with the reviewer on this point – there is a possibility that the longer "synthetic" version of Lpp is more unstable and/or degraded more readily. It’s worth noting that the transcription level of lpp was not reduced (table S6), suggesting that either: (i) the protein is less "stable" or; (ii) the protein is "targeted" for degradation in response to the stretching or; (iii) that the modified transcript is translated less efficiently.

We have clarified this in the discussion of the revised manuscript.

To summarise, I think that the study is an interesting and important contribution to the field, but that the authors need to be more conservative in the conclusions that they draw. For example, they need to provide some discussion as to the limitations of their study, and the differences with other published work (point 2).

We have now clarified and amended the discussions to better describe the limitation of the present study and attempt to outline the key differences with previous studies.

Reviewer #2:The authors addressed my concerns about the initial manuscript and submitted a substantially improved revised version. In particular, they now clarify that the 8-fold reduced cell copy number of Lpp+21 could be responsible of some of the phenotypes observed. My remaining doubt is that the observed changes in cell envelope properties (softer outer membrane, increased blebbing) and the altered essentiality of genes could be similar in cells that lack Lpp or that have 8-fold less wild-type Lpp. Overall, I am now more positive about the revised manuscript as it clarifies the question about the periplasmic widths of cells with an elongated Lpp, which has not been fully addressed in previous papers. The work by Mandela et al. provides an interesting and important analysis of how a Gram-negative bacterium maintains a largely functional cell envelope (biogenesis) upon severe disturbance in the linkage between the two key components, the peptidoglycan layer and outer membrane.

We thank the reviewer for their thoughtful assessment of our revised manuscript. We agree that we can not rule out the effect that reduction in Lpp copy number may have on the cells. In revision, we have clarified that this is a limitation of the data, but it is an important finding due to the potential implication on previous studies using the Lpp+21 isoform.

Reviewer #3:The revision by Mandela et al. addresses some but not all of the comments and suggestions that I and the other reviewers introduced in the our initial review. In particular, they have clarified the differences (or lack thereof) with Asmar et al. in terms of the periplasmic width changes due to Lpp+21. Moreover, they have helped to clarify some of the differences between their experiments and others such as the growth defect in ∆mrcB with lpp mutants, though I was surprised that they did not simply do the experiment to test whether their explanation (difference in growth media) was the case.

We have now conducted this experiment and included it in as a supplemental figure. In our synthetic screen the initial selections were done on LB media with antibiotics before testing in minimal media. Both the ΔmrcB lpp^+21^ and ΔmrcA lpp^+21^ were able to be recovered from the initial double antibiotic selection on LB and could grow on LB agar plates, albeit with an apparent minor growth defect (Figure 3—figure supplement 3).

One of my major criticisms in the initial review persists – that there are interesting conclusions that emerge from the screen of the Keio collection, but none of these results are followed up on, and hence I am left with the question of to what extent any of these phenotypes are direct. I had suggested investigating one in detail using e.g. CRISPRi knockdown in order to examine single-cell phenotypes during depletion of a now-essential protein in the lpp+21 background, but this was not done. Thus, while I find the topic of their paper generally interesting and well motivated, I am left unsatisfied with some of the major conclusions of their paper.

While we agree that it would be interesting, the proposed experiment is beyond the scope of the present study. However, it could form the basis of a subsequent manuscript. Many studies using the KEIO mutant library do not (and realistically could not) follow up on all "hits" with a conditional single cell knockdown experiments. This represents a significant amount of work, and we cannot recall any study where synthetic lethal screens were routinely investigated via such silencing methods.

We confirmed all “hits” by generating independent mutants of all relevant double knockouts.

Moreover, many of the conclusions of their paper remain to be fully justified. For instance: in their abstract:"…impacts the load-bearing capacity of the outer membrane": this is not something that they show to be true in their paper, it is an inference from another paper, and there is nothing in this work that directly addresses load bearing. I don't think they have shown adequately that the combination of tilting and reduced Lpp abundance is the cause of reduced load bearing.

We have revised the relevant text to avoid making this claim throughout.

– "E. coli homeostatically counteracts periplasmic enlargement by tilting Lpp and reducing Lpp abundance": I still don't see how there is any homeostasis here, since they are now emphasizing that Lpp+21 does in fact change periplasmic width.

We have revised the title, abstract and discussions to removed and/or toned down any reference to a "homeostatic" regulatory mechanism throughout the manuscript and have sought to place more emphasis on the cellular "adaptation" or "response" to the Lpp+21 protein.

Also, they have not shown that the reduced Lpp abundance is a cause – what happens if you increase Lpp+21 abundance via inducible expression.

In our study there was a reduction in the steady state protein level of Lpp+21 compared to Lpp. We have also demonstrated that this was not due to reduced transcription of lpp+21, indeed transcriptomic data (table S6) showed an increased level of lpp+21 transcripts.

– Line 158: note that I do not believe it has been established that stiffness of E. coli is always associated with OMV production – I would not make claims based on this hypothesized connection.

We are not suggesting that all things that effect cell stiffness affect OMV production, just that there is a documented correlation between reduced cell stiffness and OMV production. Thus we sought to investigate OMV production in our lpp mutant strain. We have clarified the relevant sentence to better represent our intent.

– Figure 2 title: there is no data here that shows that Lpp+21 cells have a softened OM.

The title to better represent the data.